# Learning to Generate Unit Tests for Automated Debugging

**Archiki Prasad**   **Elias Stengel-Eskin**   **Justin Chih-Yao Chen**
**Zaid Khan**      **Mohit Bansal**

University of North Carolina at Chapel Hill
{archiki, esteng, cychen, zaidkhan, mbansal}@cs.unc.edu

## Abstract

Unit tests (UTs) play an instrumental role in assessing code correctness as well as providing feedback to large language models (LLMs), motivating automated test generation. However, we uncover a trade-off between generating unit test inputs that *reveal errors* when given a faulty code and *correctly* predicting the unit test output without access to the gold solution. To address this trade-off, we propose UTGEN, which teaches LLMs to generate unit test inputs that reveal errors along with their correct expected outputs based on task descriptions. Since model-generated tests can provide noisy signals (e.g., from incorrectly predicted outputs), we propose UTDEBUG that (i) scales UTGEN via test-time compute to improve UT output prediction, and (ii) validates and backtracks edits based on multiple generated UTs to avoid overfitting, and helps LLMs debug effectively. We show that UTGEN outperforms other LLM-based baselines by 7.59% based on a metric measuring the presence of *both* error-revealing UT inputs and correct UT outputs. When used with UTDEBUG, we find that feedback from UTGEN's unit tests improves pass@1 accuracy of Qwen2.5 32B on HumanEvalFix and our own harder debugging split of MBPP+ by over 3.17% and 12.35% (respectively) over other LLM-based UT generation baselines. Moreover, we observe that feedback from Qwen2.5 32B-based UTGEN model can enhance debugging with frontier LLMs like GPT-4o by 13.8%. Lastly, we demonstrate that UTGEN is a better judge for code correctness, outperforming a state-of-the-art trained 8B reward model by 4.43% on HumanEval+ with best-of-10 sampling using Qwen2.5 7B. Our code and datasets are publicly available at https://github.com/archiki/UTGenDebug.

## 1 Introduction

With rapid advancements in training large language models (LLMs; Achiam et al., 2023; Anthropic, 2024; Gemini et al., 2023), enhancing their coding abilities has garnered significant attention (Chen et al., 2021; Li et al., 2022; 2023; Roziere et al., 2023; Guo et al., 2024b, *inter alia*). However, these models are far from perfect and – much like human-written code – model-written code contains errors. Human developers often improve their code through test-driven development, i.e., identifying failure cases by providing example inputs and their expected outputs – referred to as *unit tests* (UTs) – to test individual functionalities, reasoning over

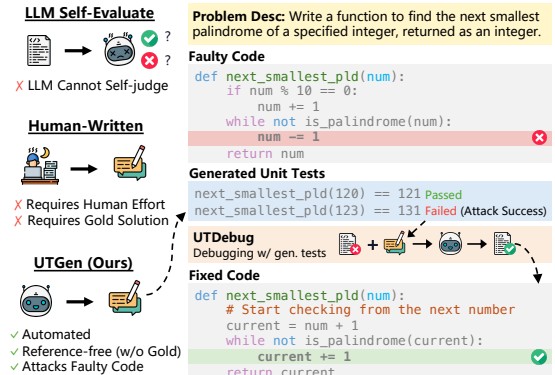

Figure 1: We propose UTGEN, which automatically generates failing unit tests (UTs) for a faulty code (triggering errors) without access to the gold solution. The generated UTs can in turn be used for LLM debugging via UTDEBUG, improving code accuracy.

causes of failure, and modifying the code to address the issues (Maximilien & Williams, 2003; Nagappan et al., 2008; Ficco et al., 2011; Beck, 2022). Models have similarly been shown to benefit from iterative debugging based on explicit or implicit feedback stemming from failing unit tests (Zhang et al., 2023; Chen et al., 2023b; Moon et al., 2023). However, the ability to provide feedback and debug incorrect code is often bottlenecked by the availability of (failing) unit tests for a given problem.

While several coding benchmarks come with human-written UTs for evaluation purposes, these suites of UTs are typically small due to the laborious annotation process (Liu et al., 2024b). Unit test collection is challenging for two reasons: first, it requires sampling inputs that are likely to trigger an error. For example in Fig. 1, the unit test: `next_smallest_pld`(120)==121 would *not* trigger any error (despite the fact that the code is incorrect for non-multiples of 10), while another unit test: `next_smallest_pld`(123)==131 would lead to an incorrect output, revealing a bug in the function. Secondly, UTs require expected outputs (e.g., 131 in the previous UT), i.e., the desired behavior of the code being tested must be known. Due to these challenges, prior work employs LLMs to generate unit test inputs at random (i.e., without conditioning on faulty code to generate failing inputs) and often uses the gold (correct) code solution for generating outputs (Chen et al., 2023a; Chae et al., 2024). Therefore, these approaches do not scale well at inference time, which requires an *online and automated* method for generating UTs as the LLMs solve problems on the fly. To bridge this gap, we pose the following research questions:

- **RQ1:** *What are desirable properties for UT generators, and how do we measure them?*
- **RQ2:** *How well do LLMs perform zero-shot UT generation, and how can we improve them?*
- **RQ3:** *How do we best use an automated but potentially noisy UT generator to improve code debugging and generation?*

To address RQ1, we characterize two desirable properties of unit test generators (in Sec. 3.1): 1) *high attack rate*, i.e., given a faulty code, the unit test generator should generate inputs that are likely to trigger errors;[1] 2) *high output accuracy*, ensuring that the UT output is consistent with the task description (and that of a correct solution). For instance, in Fig. 1, `next_smallest_pld`(120) would lead to a lower attack rate, as it does not trigger any errors, while `next_smallest_pld`(123)==131 *does* (high attack rate); however, both have high output accuracy as in both cases the output is correct. We measure these properties via three *intrinsic metrics*: measuring attack rate and output accuracy independently and then, crucially, how often a UT generator can generate *both* failing inputs along with correct UT outputs based only on the problem description and a faulty code. We benchmark the ability of LLMs to act as zero-shot unit test generators and show that coding LLMs generally struggle (cf. Sec. 5.1).

Moreover, addressing RQ2, we find that, without finetuning, zero-shot models often exhibit a strong trade-off between attack rate and output accuracy. In other words, the UTs that are most likely to trigger errors (i.e., higher attack rate) are generally more challenging edge cases, making it harder for the model to predict their output (i.e., lower output accuracy). Due to a lack of dedicated datasets for unit test generation in prior work, we introduce UTGEN, a data creation and training recipe (Sec. 3.2) that bootstraps training data for UT generation from *existing* code generation datasets by perturbing code to simulate errors, generating failing unit test and augmenting it with chain-of-thought rationales (Wei et al., 2022). We show that training via UTGEN better balances this trade-off and yields a higher number of unit tests that have *both* attacking inputs and correct outputs, with 7.59% absolute improvement over prompting models to generate failing UTs.

Finally, we examine RQ3, utilizing our generated UTs on downstream, i.e., *extrinsic* tasks: code debugging and code generation. Here, automated UT generation methods face additional challenges: unlike human-generated gold UTs – which have correct outputs but require human involvement – generated UTs provide noisy feedback, as the UT might fail to reveal the buggy code's errors or have an incorrectly predicted output. To mitigate this issue, we propose UTDEBUG, an improved multi-turn debugging method that addresses these challenges in two ways (cf. Sec. 3.3): (i) we improve the output accuracy of generated

---

[1]Here, an 'error' means either a syntax/runtime error, or when the result of the target code does not match the expected output of the UT. We expand on this in Sec. 3.

UTs by scaling test-time compute via self-consistency (Wang et al., 2022); (ii) we regularize the debugging process by generating multiple UTs and accepting code edits *only if* the revised code passes *more* generated UTs, backtracking edits otherwise. We then plug UTs generated from multiple LLM-based methods into UTDEBUG in Sec. 5.2, finding that on our most challenging subset of MBPP+ problems, UTDEBUG with UTs generated by UTGEN improves pass@1 accuracy of Qwen2.5 Coder 32B Instruct by 15.07% (absolute) compared to debugging without UTs, and by 4.61% over a zero-shot unit test generation baseline (with similar trends on 8B-scale LLMs). We demonstrate that by generating UTs with UTGEN and computing the rate of passing UTs, we can *better judge code correctness* than using scores from trained, state-of-the-art 8B reward model (RM). For best-of-10 sampling with Qwen2.5 Coder 7B Instruct, UTGEN improves accuracy by 4.43% over the RM on HumanEval+. Finally, in Sec. 5.5, we highlight the importance of high-quality unit tests generated by UTGEN when debugging with a frontier LLM like GPT-4o (Hurst et al., 2024), finding that using UTGEN-feedback from a smaller model, such as Qwen2.5 32B, significantly outperforms self-generated feedback from GPT-4o by nearly 25% (absolute) on the most challenging subset of MBPP+ problems.

## 2 Related Work

**Automatic Unit Test Generation.** Manually writing unit tests is laborious and often infeasible (Chen et al., 2023a; Liu et al., 2024b). Consequently, past research explores automatic UT generation (King, 1976; Cadar et al., 2008; Holler et al., 2012; Cha et al., 2015, *inter alia*). The advent of LLMs has spurred recent efforts in using them for UT generation (Chen et al., 2023a; Schäfer et al., 2023; Liu et al., 2024b). Specifically, Schäfer et al. (2023) and Liu et al. (2024b) focus on generating *unit test inputs* via prompting frontier LLMs like GPT-4 and/or iterative prompting, assuming access to the *gold* solution. In contrast, our models, trained with UTGEN, generate *both input-output UT pairs* based on the task description without relying on the gold implementation. While Chen et al. (2023a) also generate input-output UT pairs using standard LLM prompting, their primary focus is code generation – *not* the quality of generated UTs. In contrast, we directly model the desiderata (or quality) of UTs, and demonstrate its utility on code generation and debugging.

**LLM Debugging.** Using LLMs for debugging faulty code, or program repair, has been extensively studied. Debugging approaches are divided into those training models to debug (Moon et al., 2023; Ni et al., 2024; Chae et al., 2024) and those providing external feedback to pretrained models (Chen et al., 2023b; Zhang et al., 2023; Olausson et al., 2023; Zhong et al., 2024). Both rely on *gold* unit tests for training or feedback. Thus, UTGEN complements both methods by providing generated unit tests when human-written tests are scarce or unavailable. In Sec. 3.3, we introduce UTDEBUG, a debugging pipeline that addresses noisy feedback from inaccurate unit tests through test-time scaling and backtracking. Moreover, in Sec. 5 we show that UTGEN's unit tests can effectively provide feedback to LLMs for code generation and debugging.

## 3 Unit Test Generation and Automated Debugging

### 3.1 Background and Task Setup

Given a natural language task description $d$ for a coding task, we focus on generating unit tests (UTs) in the format of *input-output pairs* that are consistent with the task description $d$. Our setup is consistent with Chen et al. (2023a) and Jain et al. (2024) who also consider unit tests in the form of input-output pairs generated *without* utilizing the correct implementation of the function. More generally, our setting is akin to parameterized unit testing (Tillmann et al., 2010) and uses the notion of *functional correctness*, i.e., measuring correctness by simulating an exhaustive set of scenarios (UT inputs) and ensuring that the function performs as expected (as per the problem description $d$).

**Notation and Desiderata.** Let $d$ denote the natural language description of the function to be implemented (top yellow box in Fig. 1). We assume that this description specifies the

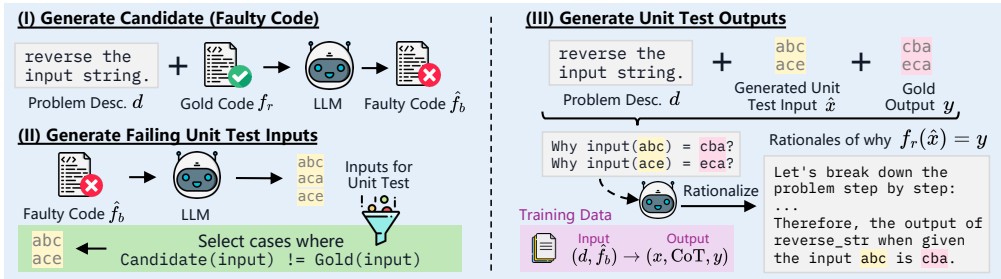

Figure 2: **UTGEN Training Pipeline:** Starting with training data for code generation (problem description and gold code), we create training data for UT generation in three stages: (I) perturbing gold code to generate faulty codes, (II) generating UT inputs and filtering for failing UTs, and (III) generating and relabeling chain-of-thought rationales conditioned on the gold code's outputs.

input space $\mathcal{X}$ and output space $\mathcal{Y}$. Furthermore, let the set of all functions that correctly solve the task be $\mathcal{F}_d$. Then, a *valid* unit test $(x, y)$ for task $d$ is:

- $x \in \mathcal{X}$, i.e., the input of the unit test is a *valid input* as per the task description $d$.
- $f_r(x) = y, \forall f_r \in \mathcal{F}_d$, i.e., $y$ is the *expected output* as per the task description $d$, and therefore, is the result one gets by executing any correct implementation.

For example, in Fig. 1, 120 is a valid input, as it is a number, whereas "apple" is not. Similarly, 121 is the expected output of the function (given 120 as input), while 122 would be an invalid output. Thus, (120, 121) is a valid unit test for the task.

**Unit Test Generator.** Addressing RQ1, we define the desirable properties of an automatic unit test generator $T_\theta$, parameterized by $\theta$. Ideally, $T_\theta$ should generate *valid unit tests* from a task description $d$, i.e., $T_\theta(d) \mapsto (x, y)$ without any manual intervention. However, to account for downstream utility of unit tests, we denote a potentially buggy code under testing as $\widehat{f}_b$. If $\widehat{f}_b \notin \mathcal{F}_d$ is faulty or implemented incorrectly, a *desirable* unit test generator should be able to efficiently generate *failing unit tests*: $T_\theta(d, \widehat{f}_b) \mapsto (x, y)$, such that $\widehat{f}_b(x) \neq y$, i.e., a valid unit test input $x$ that uncovers the errors in $\widehat{f}_b$. Moreover, $T_\theta$ must also predict the correct UT output $y$, i.e., the output of the correct code. We study generators of the form $T_\theta(d)$, which consider only the description, and debugging-style generators of the form $T_\theta(d, \widehat{f}_b)$, which also consider a buggy code solution $\widehat{f}_b$. Empirically, we find that $T_\theta(d)$ lacks sufficient context to generate effective error-revealing tests. Therefore, we focus on training UT generators of the form $T_\theta(d, \widehat{f}_b)$.

### 3.2 UTGEN: Training LLMs for Unit Test Generation

While prior work focuses on curating training data for improving code generation (Guo et al., 2024a; Muennighoff et al., 2023), there is a general lack of dedicated datasets for training the desired UT generator outlined above. Therefore, to improve along the lines of RQ2, we design, UTGEN, a training recipe that bootstraps this data from training datasets for code generation i.e., a collection of problem descriptions and their corresponding code.

**Problem Descriptions and Target Codes.** We start with a collection of coding problems with problem descriptions ($d$) and gold codes ($f_r$) as illustrated in Fig. 2 (I). Specifically, we use publicly available data from Tulu-3 (Lambert et al., 2024a) due to its large scale and the improvements noted by Lambert et al. (2024a) when finetuning LLMs on coding tasks.[2] We filter it to focus on Python code with functional abstractions (further details in Appendix B). However, in order to train a unit test generator we would need access to incorrect or faulty

---

[2]The code solutions in the Tulu-3 SFT mix have either been written by humans or by frontier models and are thus highly likely to be correct (Lambert et al., 2024a).

code that can be debugged, as unit tests must have some error to attack. We obtain these by using the LLM to perturb the reference code solution.

**Annotating Unit Tests.** As mentioned in Sec. 3.1, one of the goals of the unit test generator is to be able to not only generate valid unit tests, but *failing* unit tests (that trigger errors). To facilitate this, given a problem description, reference code $f_r$, and buggy candidate code $\widehat{f}_b$, we sample $n$ different unit test inputs (via the prompts in Appendix H). A unit test $(x, f_r(x))$ is *failing* if $f_r(x) \neq \widehat{f}_b(x)$, i.e., if the output of the candidate fails to match the reference output (cf. Fig. 2 (II)). Note that, while the LLM can be used to generate the output of unit test, it can often be inaccurate (Jain et al., 2024; Gu et al., 2024); so to ensure output accuracy, we use the output of the reference code during training. Due to the formalization of UT output prediction as a *reasoning word problem*, we further explore generating chain-of-thought (CoT; Wei et al., 2022) reasoning before predicting the UT output. To this end, we employ the post-hoc rationalization procedure outlined in Zelikman et al. (2022) – given the entire UT $(x, f_r(x))$, we ask the LLM to generate rationales supporting why $f_r(x)$ is the output corresponding to the input $x$. Then, to create the supervision data, we add these rationales as CoTs prior to the output prediction, as illustrated in Fig. 2 (III). We revisit the discussion of how accurate LLMs are at output prediction during inference in Sec. 5.

**Supervised Finetuning.** When training LLMs, the input to the LLM is the same prompt used for sampling unit tests (listed in Appendix H) and the output is a failing unit test. We collect nearly 30K training instances for 8B-scale models and roughly 70K instances for training 32B LLMs (details in Appendix B). The goal here is to improve both output accuracy and attack rate *jointly* via supervised finetuning using the negative log-likelihood loss using basic hyperparameters outlined in Appendix B.

### 3.3 UTDEBUG: Generated Unit Tests for Debugging

A key difference when using generated unit tests, as opposed to human-generated UTs, is the *degree of noise* in the feedback. Despite training models via UTGEN or other methods, a generated unit test $(\widehat{x}, \widehat{y})$ may not be 100% accurate. This can manifest in two ways: 1) the generated UT *input does not fail* for the code under debugging $\widehat{f}_b$, i.e., $f_r(\widehat{x}) = \widehat{f}_b(\widehat{x})$; 2) the generated UT *output is inaccurate*, i.e., not consistent with what a gold solution would yield ($\widehat{y} \neq f_r(\widehat{x})$). Both types of errors can negatively impact the utility of unit tests for debugging, as shown in Fig. 3 (left). A non-adversarial input might result in faulty code being misclassified as correct and prematurely removed from debugging. Additionally, incorrect outputs can cause false positives, introducing errors to otherwise correct code. Even if the candidate code has bugs, incorrect UT outputs can lead to incorrect feedback, thereby, degrading performance. These issues motivate our RQ3 to incorporate noisy feedback. We propose UTDEBUG, which includes two ways to mitigate noisy feedback from automatically generated unit tests (additional details in Appendix C). We empirically validate the importance of both these components of UTDEBUG in Appendix E.

**Boosting Output Accuracy via Test-Time Scaling.** Building on past work that has shown the benefits of scaling up inference-time compute (Wang et al., 2022; Lightman et al., 2023; Snell et al., 2024), we improve UT output accuracy by allocating additional computation to the problem. Specifically, we use self-consistency (SC; Wang et al., 2022), whereby, for a given UT input, we sample $k = 8$ output completions (including CoT rationales) and take the most common final UT output (majority vote) as the final answer, as shown in Fig. 3 (top-right). To further boost output accuracy, consistent with Prasad et al. (2024), we upsample UT inputs and only retain those where the final answer gets over 50% of the votes (i.e., 4 votes), discarding the unit test otherwise.

**Back-Tracking and Validation.** To handle noisy feedback, it is crucial to know when to *abstain or backtrack*, e.g., discard edits. This is useful when generated UT outputs are incorrect, leading to faulty feedback, or when correct feedback is not incorporated by the model. Inspired by test-driven development, we accept changes only if the revised code *passes the previously failing UT*. Moreover, developers often use multiple UTs to detect errors from changes in other parts of the code. This helps prevent *overfitting*, where code is

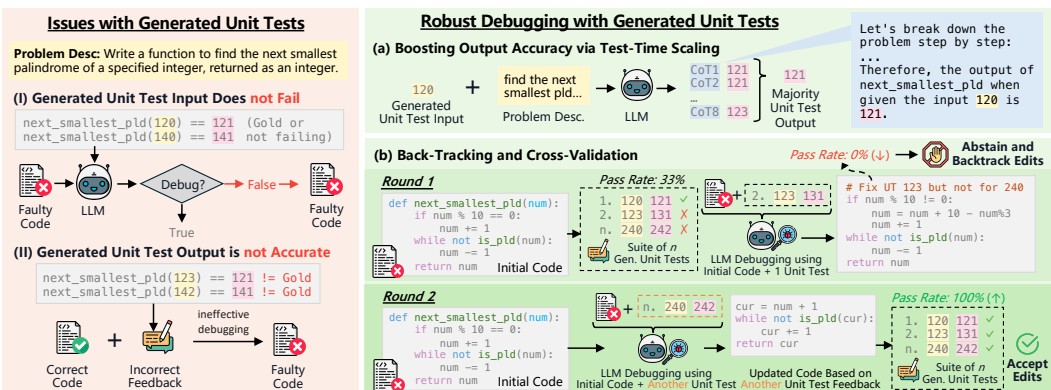

Figure 3: **Left**: We highlight potential issues with debugging a faulty code using generated UTs: (I) *non-failing* UTs misclassify faulty code as correct; (II) UTs with incorrect outputs produce incorrect feedback and consequently, unsuccessful debugging. **Right**: We introduce UTDEBUG which (a) uses inference-time scaling to select better UT outputs based on a majority vote, and (b) generates multiple UTs for validation, discarding edits when overall pass rate decreases (round 1) and accepting edits when overall pass rate improves (round 2).

modified to pass a single unit test but fails others. Therefore, in each debugging round via UTGEN, we generate *n* UTs. We use one for debugging feedback and accept edits only if the pass rate on the entire set improves (validation); otherwise, we backtrack. This process is shown in Fig. 3 (b), where edits in round 1 are discarded, but those in round 2 improve the test suite as a whole and are accepted.

# 4 Experimental Setup

**Models.** We demonstrate the effectiveness of UTGEN on three 7-8B scale LLMs across different model families that are adept at coding, namely, Llama3 8B Instruct (AI@Meta, 2024), Llama3.1 8B Instruct (Dubey et al., 2024), Qwen 2.5 Coder 7B Instruct in addition to the stronger Qwen 2.5 Coder 32B Instruct model (Hui et al., 2024).

**Datasets.** We use debugging datasets based on popular LLM coding benchmarks, HumanEval (Chen et al., 2021) and MBPP (Austin et al., 2021) along with their extended evaluation suites with more unit tests as proposed by (Liu et al., 2024b). We describe their construction below, with further details in Appendix A.1.

- **HE+Fix.** We take HumanEvalFix (Muennighoff et al., 2024), which has human-introduced errors, and following Ni et al. (2024), augment each problem's test suite with overlapping problems from EvalPlus (Liu et al., 2024b). This yields 158 problems, each with a faulty solution and a suite of private UTs for evaluation – i.e., UTs not used for debugging.
- **MBPP+Fix.** We construct a debugging dataset based on the MBPP+ benchmark (Austin et al., 2021; Liu et al., 2024b; Ni et al., 2024). To obtain faulty codes, we sample 16 solutions for each problem from different 7-8B scale models and filter for incorrect solutions based on the private unit tests (described in Appendix A.1). Then, we sample one faulty solution for each problem, resulting in 325 problems with realistic incorrect solutions generated by LLMs. We find such errors are generally harder for models to debug (cf. Sec. 5).
- **MBPP+Fix (Hard).** Additionally, we create a different split from MBPP+ with more subtle errors, logical flaws and missing corner cases making it harder to debug. To this end, we follow a similar generation and filtering process as above but only retain faulty code that passes between 50% - 95% of unit tests, resulting in 170 problems with faulty codes.

**Evaluation Metrics.** Below we describe three intrinsic evaluation metrics for unit test generation (with additional details in Appendix A.2):

- **Attack Rate.** We measure the attacking ability of a UT generator by the frequency with which the output of a gold solution $f_r$ for input $\hat{x}$ differs from that of the buggy code $\hat{f}_b$.

- **Output Accuracy.** Measures how often the output of a generated unit test $\widehat{y}$ is consistent with the problem description, i.e., generates the same output as the reference code $f_r$.
- **Accuracy ∩ Attack.** This metric combines both attack rate and output accuracy and represents how often a unit test generator $T_\theta$ generates a useful (i.e., *failing*) unit test for a given target code $\widehat{f_b}$ while *also* predicting the output *correctly*.

We rely on intrinsic metrics during development on a validation split from MBPP+ data, e.g., for checkpoint selection and prompt design. Then, we measure the utility and effectiveness of UTGEN by computing the **pass@1 code accuracy**, i.e., the percentage of codes passing *all* unit tests on code debugging and generation tasks.

**Baselines.** We compare UTGEN against the following (prompts in Appendix H):

- **No UT feedback:** In this baseline, we use self-generated or self-critique feedback, following prior work (Madaan et al., 2024; Shinn et al., 2024; Chen et al., 2023b). Specifically, we prompt the model to generate an explanation about the bugs in the target code in addition to determining if the target code is correct, i.e., does not have any more bugs.
- **Randomly-sampled UTs:** Consistent with Chen et al. (2023a), we relax the requirement for the LLM to generate failing unit tests, and prompt the model to jointly generate valid unit tests (inputs and outputs) based on the task description, irrespective of the target code, i.e., we sample $(\widehat{x}, \widehat{y}) \sim T_\theta(d)$. We further expand on the comparison with Chen et al. (2023a) in Appendix E.
- **Prompted Failing UTs:** Using the same prompts as UTGEN without any training, we prompt an LLM to generate a failing UT (with a correct output) given the description and the target code. Here, we sample $(\widehat{x}, \widehat{y}) \sim T_\theta(d, \widehat{f_b})$.

Note that all UT-generation baselines along with UTGEN utilize the test-time scaling and backtracking approaches of UTDEBUG highlighted in Sec. 3.3 and exhibit similar run times.

## 5 Results and Analysis

### 5.1 Intrinsic Evaluation of UT Generation

In order to study the inherent UT generation abilities of different models and baselines, we use the intrinsic metrics defined by RQ1 and outlined in Sec. 4 in Table 1 on the most challenging debugging task MBPP+Fix (Hard) averaged over 3 runs.

**Trade-offs in Attack Rate and Output Accuracy.** In Table 1, we benchmark the zero-shot abilities of different models on UT generation, as well as our improved training method (corresponding to RQ2). Here, we observe a clear tradeoff: *prompted and random unit test generation each optimizing output accuracy or attack rate at the cost of the other metric*. While randomly sampled UTs have relatively higher output accuracy (i.e., the output of the unit test is correct according to the problem description), the random baseline often lacks the ability to generate UTs with inputs that trigger errors (failing UTs). This can be explained by the fact that it is not conditioned

| Model | Method | Attack Rate | Out Acc. | Acc. ∩ Attack |
|---|---|---|---|---|
| Llama3 8B | Random | 30.98 | 46.86 | 10.24 |
| | Prompted | 38.04 | 37.10 | 11.51 |
| | UTGEN | **41.18** | **48.24** | **16.57** |
| Llama3.1 8B | Random | 21.76 | 39.80 | 9.02 |
| | Prompted | 40.59 | 26.86 | 9.61 |
| | UTGEN | **41.37** | **47.75** | **16.67** |
| Qwen2.5 7B | Random | 26.27 | 57.25 | 13.53 |
| | Prompted | 39.80 | 40.78 | 14.71 |
| | UTGEN | **41.96** | **58.04** | **20.20** |
| Qwen2.5 32B | Random | 25.29 | 47.25 | 11.37 |
| | Prompted | 51.96 | 48.43 | 29.41 |
| | UTGEN | **56.08** | **59.22** | **34.71** |

Table 1: **Evaluation on intrinsic metrics** of different UT generation baselines and UTGEN across 7-8B different model families on MBPP+Fix (Hard) over 3 runs. Higher is better for all three intrinsic metrics.

on the faulty code $\widehat{f_b}$. On the other hand, when models are prompted to generate a UT that breaks the faulty code in the prompted UT baseline, they generally achieve a higher attack rate; however, they lag in terms of output accuracy. For instance, in the case of Qwen2.5 7B, switching from random sampling to prompting failing UTs improves the attack rate by

13.53% but decreases the output accuracy by 16.47%, hence leading to comparable Acc. ∩ Attack scores (13.53% vs 14.71%). Taken together, this suggests the presence of competing desirable traits of UT generators (cf. Sec. 3.1; RQ1): without training, one baseline often generates trivial unit tests for which the output can easily be predicted (high output accuracy, low attack rate), while the other generates challenging unit tests for which it cannot produce the expected answer (low output accuracy, high attack rate). It also indicates that while models can generate challenging unit test inputs that result in code failure, *models struggle with reasoning over correct outputs of failing UT inputs*.

**UTGEN is Most Effective at Unit Test Generation.** We propose to break through this trade-off (thereby addressing RQ2, *"How can we improve LLMs' UT generation abilities?"*) by training models with UTGEN, where we directly supervise models to have both high output accuracy and high attack rate. Ideally, UTGEN should lead to models that generate unit tests in the "sweet spot" of difficulty, where they are hard enough to trigger errors but not so hard that their output cannot be predicted. Table 1 shows that models trained with UTGEN consistently rank highest at jointly generating *failing UT inputs with correct outputs* (as measured via Acc. ∩ Attack in Table 1). For instance, on the strongest coding model in our setting, Qwen2.5 32B, UTGEN obtains the highest attack rate, and output accuracy as well as improves Acc. ∩ Attack score by 5.3% (absolute) over failing UTs generated by prompting and by 23.34% over randomly generated UTs. Similarly, Llama3.1 improves (in terms of attack rate) over randomly-sampled UTs by 19.61% and the output accuracy of prompted failing UTs by 20.89%, ultimately improving the joint Acc. ∩ Attack score by up to 7.65%. We report the intrinsic metrics for HE+Fix and MBPP+Fix in Appendix D.

## 5.2 Generated UTs for Code Debugging

**UTs from UTGEN are Best for UTDEBUG.** Addressing RQ3, we measure how effective each type of UT generator is in a downstream debugging evaluation. From Table 2, we observe that across different LLM families, multi-turn debugging with UTDEBUG is more effective when using generated UTs from UTGEN than debugging with UTs generated by the baselines, as well as debugging without feedback. For instance, with Qwen2.5 32B, after 3 rounds of debugging on MBPP+Fix, UTGEN improves over debugging with randomly-sampled UTs by 4.61%, debugging by failing UTs by 3.38%, and debugging without UT feedback by 15.07%. Moreover, on the more challenging MBPP+Fix (Hard) split, UTGEN improves over randomly-sampled UTs by 22.35% and the baseline without UT feedback by 12.35%. Given that we use the same underlying LLM and similar feedback templates, the results in Table 2 show that UTs generated by UTGEN provide the *most useful and effective* feedback. Lastly, we find that LLM debugging without any UT feedback is least effective across model families, thus, establishing that despite noise and issues discussed in Sec. 3.3 and Fig. 3, UTDE-BUG with UT-based feedback is more effec-

| Model | UT Method | HE+Fix | MBPP+Fix | (Hard) |
|---|---|---|---|---|
| Llama3 8B | No UT | 27.22 | 16.31 | 11.76 |
| | Random | 51.90 | 30.46 | 17.06 |
| | Prompted | 51.90 | 28.92 | 22.94 |
| | UTGEN | **53.80** | **37.54** | **28.82** |
| Llama3.1 8B | No UT | 31.65 | 10.15 | 11.18 |
| | Random | 62.03 | 33.54 | 13.53 |
| | Prompted | 56.33 | 28.00 | 24.71 |
| | UTGEN | **67.09** | **36.92** | **28.23** |
| Qwen2.5 7B | No UT | 52.53 | 23.08 | 16.47 |
| | Random | 79.75 | 34.77 | 17.06 |
| | Prompted | 75.32 | 32.92 | 24.12 |
| | UTGEN | **82.91** | **37.54** | **29.41** |
| Qwen2.5 32B | No UT | 79.11 | 39.08 | 32.94 |
| | Random | 84.81 | 49.54 | 22.94 |
| | Prompted | 85.44 | 50.77 | 40.59 |
| | UTGEN | **88.61** | **54.15** | **45.29** |

Table 2: **Evaluating pass@1 accuracies after debugging with UTDEBUG**, using UTs generated by UTGEN and other baselines for 3 rounds, on HumanEval+Fix (HE+Fix), MPBB+Fix, and MBPP+Fix (Hard).

tive than self-generated feedback. This aligns with past work indicating that LLMs are poor at self-critiquing (Huang et al., 2024).

**Debugging Difficulty Varies Across Datasets.** Comparing the post-debugging accuracy across the three datasets in Table 2 suggests different difficulty levels when it comes to LLM debugging. The relatively high accuracies on HE+Fix (especially with Qwen2.5) suggest that human-introduced errors in that dataset are obvious and thus too easy for the models to

discover and debug. This is also corroborated by the fact that starting codes in this dataset have a pass rate of 0%, i.e., fail on *all* unit tests, suggesting high-level coarse-grained errors, e.g., syntax errors that prevent execution, or incorrect function references. On the other hand, MBPP+Fix (Hard) (with an initial pass rate of $\approx$75%) appears to be the hardest to debug, with the lowest overall post-debugging accuracy across models – up to a difference of 53.5% in final accuracies of HumanEvalFix and MBPP+Fix (Hard) for UTGEN based on Qwen2.5 7B. This in turn suggests that LLMs still struggle with identifying and fixing subtle flaws in generated code, especially in scenarios involving corner cases wherein the model passes some unit tests and fails at the rest.

## 5.3 Generated UTs for Code Generation

We further demonstrate the effectiveness of generated UTs at judging code correctness via Best-of-$N$ sampling ($N = 10$). Given a problem $d$, we sample $N$ different code solutions, and for each solution, we generate 3 UTs (via UTGEN and baselines in Sec. 4 with test-time scaling). In the end, we collate all generated UTs from each method (removing any duplicates) and choose the generated code solution that passes the most generated UTs. In other words, we use UTGEN to rerank 10 generated solutions. In addition to comparing to the random and prompt UT generation baselines, we also rerank according to a state-of-the-art 8B scale reward model (Liu et al., 2024a) on HE+ and MBPP+ (two standard code-generation datasets). Results in Table 3 reveal that:

| Model | Judge Method | HE+ | MBPP+ |
|---|---|---|---|
| Llama3 8B | -† | 54.43 | 56.00 |
| | RM (Liu et al., 2024a) | 60.76 | 59.47 |
| | Random UT | 61.39 | 58.93 |
| | Prompted UT | 63.29 | 60.00 |
| | UTGEN | **65.12** | **61.60** |
| Qwen2.5 7B | -† | 81.64 | 65.06 |
| | RM (Liu et al., 2024a) | 83.54 | 66.67 |
| | Random UT | 85.44 | 68.80 |
| | Prompted UT | 85.71 | 67.47 |
| | UTGEN | **87.97** | **70.13** |

Table 3: **Best-of-10 code accuracy after judging code correctness** via generated UTs and external reward model (RM). †Code accuracy after sampling 1 solution.

- **Generated UTs Outperform External RMs.** In all four settings (models and datasets) in Table 3, we find that LLM-generated UT are better judges for selecting code solutions with higher pass rates as compared to trained, state-of-the-art 8B RM.
- **UTGEN is a Better Judge for Code.** Across models and datasets, using UTs from UTGEN to select code yields the highest performance, outperforming external RM by 4.36%, randomly generated UTs as employed by Chen et al. (2023a) by 3.73%, and prompted UTs by 1.83% on HE+ with Llama3 8B (with similar trends on MBPP+ and using Qwen2.5 7B).

## 5.4 Comparison to Unit Test Generation by Frontier LLMs

In Table 1 we evaluated open-source LLMs that we trained using UTGEN, and in Sec. 3.3 we applied these models in our UTDEBUG framework for debugging, finding the trained models to be effective. Table 4 compares stronger frontier models – GPT-4o (Hurst et al., 2024) and DeepSeek-V3 (Guo et al., 2024b) – to the best open-source model tested (Qwen2.5 32B). Here, we see that even these stronger and much larger frontier models

| Model | Method | Attack Rate | Out Acc. | Acc. ∩ Attack |
|---|---|---|---|---|
| GPT-4o | Random | 23.92 | 57.25 | 17.25 |
| | Prompted | 54.51 | **60.39** | 33.33 |
| DeepSeek V3 | Random | 24.31 | 63.53 | 17.84 |
| | Prompted | 54.31 | 58.04 | 33.33 |
| Qwen2.5 32B | Random | 25.29 | 47.25 | 11.37 |
| | Prompted | 51.96 | 48.43 | 29.41 |
| | UTGEN | **56.08** | 59.22 | **34.71** |

Table 4: Evaluation of frontier models on intrinsic metrics on MBPP+Fix (Hard) over 3 runs.

struggle at the task of generating failing unit tests. First, we find that when prompting models with faulty codes, the attack rate of the generated UTs is slightly over 50% on MBPP+Fix (Hard) dataset, showcasing the inherently challenging nature of isolating corner cases and subtle errors in partially correct code. Furthermore, UTGEN beats both GPT-4o and DeepSeek-V3 in terms of Acc. ∩ Attack, with a gain of 1.38% over both, as well as in terms of Attack Rate. On output accuracy, UTGEN is better than DeepSeek-V3 and slightly worse than GPT-4o. Overall, we find that using UT-based feedback even with frontier LLMs

only triggers and identifies errors 1 in 3 times, leaving tremendous room for growth in the ability of models to identify partially incorrect code.

Note that cost is a major factor in automated debugging; we show that using UTGEN for debugging in UTDEBUG is successful, but involves generating multiple UTs with self-consistency across several rounds of debugging for each problem (cf. Algorithms 1 and 2). Such calls will quickly become costly on frontier models like GPT-4o and DeepSeek-V3, making relatively smaller models trained with UTGEN more attractive options.

### 5.5 Importance of Unit Tests with a Frontier Debugger

Finally, we study the importance of unit tests when using a more powerful, frontier model for debugging. To this end, we use GPT-4o (Hurst et al., 2024) as the debugger and evaluate its performance on the MBPP+Fix (Hard) benchmark. We compare a baseline that *does not* use UT-execution feedback, where GPT-4o generates its own feedback, or uses feedback generated by Qwen2.5 32B (without any unit tests) against scenarios where it uses UT feedback from the smaller Qwen2.5 32B model, both from a prompted baseline and from our proposed UTGEN. The results, presented in Table 5, show that even with a significantly stronger debugger model, feedback from *high-quality unit tests remains crucial* for effective debugging – both the prompted baseline and UTGEN outperform the "no UT" baselines by 11.17% and ≈25% (absolute), respectively. This demonstrates not only that UT feedback is vital even for frontier models, but also that the quality of the unit tests is a key factor, as UTGEN outperforms the prompted baseline by a large margin of 13.8%.

| Debugger | UT Method (Model) | MBPP+Fix (Hard) |
|---|---|---|
| GPT4o | No UT (GPT-4o) | 34.71 |
| | No UT (Qwen 32B) | 32.94 |
| | Prompted (Qwen 32B) | 45.88 |
| | UTGen (Qwen 32B) | **59.69** |

Table 5: Performance of GPT-4o as a debugger on MBPP+Fix (Hard) with different UT-generation methods. High-quality UTs from UTGEN significantly improve the performance of GPT-4o.

## 6 Conclusion

We first identified a key trade-off between attack rate and output prediction accuracy when predicting unit tests with models. In light of this trade-off, we introduced UTGEN, a new method for creating training data and teaching models to produce unit tests. This allows us to train models to produce better unit tests, as measured by intrinsic metrics like attack rate and output accuracy. Moreover, finding that existing data contains large numbers of easy errors, we introduce a new subset of data with challenging and hard-to-diagnose errors. To enable debugging with automated unit tests, we propose UTDEBUG, wherein we augment our predictor's accuracy with test-time scaling and regularize it using a cross-validation and back-tracking procedure that prevents it from overfitting to a narrow or incorrect unit test. Additionally, in Appendix F, we demonstrate that these gains persist as we scale the number of generated UTs across datasets. This, combined with UTGEN, results in consistent increases in debugging performance across models, and can be used to improve code generation via best-of-N sampling. Finally, we note that UTGEN is complementary to work on handling real-world programming issues such as Jimenez et al. (2024), which focus on debugging GitHub issues raised by people; in the future, such issues could be identified by a unit test generator such as UTGEN.

## Acknowledgments

We thank Swarnadeep Saha for insightful discussions in the early phases of this work. Furthermore, we thank the anonymous reviewers and area chairs for their helpful feedback throughout the reviewing process. This work was supported by NSF-CAREER Award 1846185, DARPA ECOLE Program No. HR00112390060, NSF-AI Engage Institute DRL-2112635, and the Center for AI Safety Compute Cluster. Any opinions, findings, and conclusions or recommendations in this work are those of the author(s) and do not necessarily reflect the views of the sponsors.

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

## A  Experimental Setup Details

### A.1  Debugging Datasets

We use debugging datasets based on popular LLM coding benchmarks, HumanEval (Chen et al., 2021) and MBPP (Austin et al., 2021) along with their extended evaluation suites with more unit tests as proposed by (Liu et al., 2024b).

- **HE+Fix.** We start with the HumanEvalFix dataset containing human-introduced errors to gold solutions proposed by Muennighoff et al. (2024) (released under MIT license). However, this dataset only uses minimal unit tests to evaluate code correctness (from HumanEval) which has shown to be unreliable (Li et al., 2023), as it can miss errors due to low coverage. Therefore, we filter for problems that overlap with EvalPlus (Liu et al., 2024b) released under Apache-2.0 license, which contains over $80\times$ more unit tests. This yields 158 problems that each have a faulty solution and an expanded suite of private UTs – i.e., UTs not used for debugging – for evaluation.

- **MBPP+Fix.** We construct a debugging dataset based on the MBPP+ benchmark (Austin et al., 2021; Liu et al., 2024b; Ni et al., 2024). To obtain faulty codes, we sample 16 solutions for each problem from different 7-8B scale models and filter for incorrect solutions based on the private unit tests (described in Appendix A.1). Then, we select one faulty solution at random corresponding to each problem. This results in a total of 325 problems with incorrect solutions representing realistic errors made by LLM coders. We find such errors are generally more challenging for models to debug (cf. Sec. 5).

- **MBPP+Fix (Hard).** To make the debugging task more challenging, we create a different split from MBPP+ with more subtle errors that are harder to debug. We identify these subtle errors by following a similar code generation setup described above, but only retain faulty code that passes between 50% - 95% of unit tests, as these partially correct solutions contain less obvious logical flaws and often require handling difficult corner cases. This results in a total of 170 problems with faulty codes.

**HE+Fix.** This dataset contains a total of 158 problems each with one incorrect human-written code and has an initial pass rate (prior to debugging) of 0%, i.e., all private unit tests are failing. As mentioned in Sec. 4, we use the dataset provided by Muennighoff et al. (2024)[3] but replace the test set for each problem from the original test suite in HumanEval (Chen et al., 2021) to that in the EvalPlus evaluation suite (Liu et al., 2024b). This increases the average unit tests per problem from 8.17 to 775.18 gold unit tests. Note that we have an automatic UT extraction script for the test code in EvalPlus, and we only retain problems for which this extraction is successful (158 out of 164).

**MBPP+Fix and MBPP+Fix (Hard).** We begin with 378 problems in the MBPP+ dataset (Liu et al., 2024b)[4] and follow the same gold UT extraction step described above, discarding problems for which the extraction fails. This leaves us with 375 problems, for which we sample 16 solutions per problem across multiple LLMs: Gemma-7B-IT (Team et al., 2024), Llama3 8B Instruct (AI@Meta, 2024), Llama3.1 8B Instruct (Dubey et al., 2024), Tulu-3 8B SFT (Lambert et al., 2024a), DeepSeek 7B coder (Guo et al., 2024b), Qwen2.5 Coder 7B (Hui et al., 2024). To generate MBPP+Fix, we filter for incorrect solutions (i.e., with at least one failing UT) and then randomly sample one incorrect code per problem. This yields 325 problems in total, each with one faulty code. This dataset has an initial pass rate of 24.21% and an average of 107.45 gold unit tests per problem. In order to construct, MBPP+Fix (Hard), we follow a similar process but select only incorrect solutions which pass 50 - 95% of unit tests. The intuition here is that solutions that are *partially* correct are often harder to debug than those that are fully incorrect. We then randomly sample one such incorrect solution per problem, yielding a dataset of 170 problems with an initial pass rate of 74.83% and an average of 107.49 gold unit tests.

**Code Generation.** When using generated UTs for improving code generation, we evaluate on HumanEval+ (HE+) and MBPP+ benchmarks proposed in Liu et al. (2024b) released under Apache-2.0 license. As described above, we filter out problems for which we can extract private UTs, yielding 158 problems in HE+ dataset and 375 problems in MBPP+ dataset. For the RM baseline we use the latest version of the 8B model based on Llama-3.1 backbone in Liu et al. (2024a).[5] We made this choice based on on rankings on the RewardBench leaderboard Lambert et al. (2024b)

## A.2 Evaluation Metrics

Below we describe three intrinsic evaluation metrics for unit test generation: *attack rate, output accuracy,* and *accuracy ∩ attack*; along with pass@1 accuracy as the extrinsic metric to measure LLM's debugging abilities using UT-feedback.

- **Attack Rate.** This metric measures a UT generator $T_\theta$'s attacking ability, i.e., its ability to generate a failing unit test input $\widehat{x}$ for a given buggy solution $\widehat{f_b}$. We measure this by matching if the output of a gold reference solution $f_r$ for input $\widehat{x}$ differs from that of the buggy code $\widehat{f_b}$. Note that this does not take into account the accuracy of the unit test output which we measure separately below. Mathematically, for any dataset $\mathcal{D}$ of coding problems, attack rate is defined as:

$$\text{AttackRate} = \frac{100}{|\mathcal{D}|} \times \sum_{d \in \mathcal{D}} \mathbb{1}_{f_r(\widehat{x}) \neq \widehat{f_b}(\widehat{x})};$$
$$\text{where}(\widehat{x}, \widehat{y}) \sim T_\theta(d, \widehat{f_b})$$

- **Output Accuracy.** This metric measures how often the output of a generated unit test $\widehat{y}$ is consistent with the problem description, i.e., generates the same output as the reference gold code $f_r$. Output accuracy does not require the generated unit test to fail. In other words,

---

[3]https://huggingface.co/datasets/bigcode/humanevalpack
[4]https://huggingface.co/datasets/evalplus/mbppplus
[5]https://huggingface.co/Skywork/Skywork-Reward-Llama-3.1-8B-v0.2

---

**Algorithm 1** `BuildUT`: Build Generated Unit Test Suite

---

**Input:** $d, \hat{f}_b$ // *problem description and buggy code*
**Params:** Number of UTs $n$, Number of SC samples $k$, Unit Test Generator $T_\theta$
**Output:** Set of Generated UTs $\mathcal{U}$
$\mathcal{U} \leftarrow \varnothing$ // *Initialization, i.e., no generated UTs*
**for** UT index $i \in [1, \cdots, 3 \times n]$ **do**
    // *Generates up to n distinct UTs from the UT generator $T_\theta$*
    $(\hat{x}^i, \hat{y}^i) \sim T_\theta(d, \hat{f}_b)$ // *Sample UTs from UT generator*
    $v^i \leftarrow \varnothing$ // *Initialize vote lookup for self-consistency*
    **for** Output index $j \in [1, \cdots, k]$ **do**
        $r^i_j \sim T_\theta(d, \hat{f}_b | x^i)$ // *Sample UT output*
        $y_j \leftarrow \texttt{extractAns}(\texttt{r}^{\texttt{i}}_{\texttt{j}})$ // *Extract UT output*
        $v^i[y_j] \leftarrow v^i[y_j] + 1$ // *Append vote tally*
    $\hat{y}^i \leftarrow \arg\max(v^i)$ // *Use majority vote for UT output*
    **if** $v^i[\hat{y}^i] \geq 0.5k$ **then** // *Answer gets over 50% of the vote*
        $\mathcal{U} \leftarrow \mathcal{U} + (\hat{x}^i, \hat{y}^i)$ // *Add to generated UT set*
    **if** $|\mathcal{U}| \geq n$ **then return** $\mathcal{U}$
**return** $\mathcal{U}$

---

$$\text{OutputAcc} = \frac{100}{|\mathcal{D}|} \times \sum_{d \in \mathcal{D}} \mathbb{1}_{\hat{y} = f_r(\hat{x})};$$
$$\text{where} (\hat{x}, \hat{y}) \sim T_\theta(d, \hat{f}_b)$$

- **Accuracy $\cap$ Attack.** This metric combines both attack rate and output accuracy and represents how often a unit test generator $T_\theta$ generates a useful (i.e., *failing*) unit test for a given target code $\hat{f}_b$ while *also* predicting the output *correctly*. We calculate this as,

$$\text{Acc.} \cap \text{Attack} = \frac{100}{|\mathcal{D}|} \times \sum_{d \in \mathcal{D}} \mathbb{1}_{f_r(\hat{x}) \neq (\hat{f}_b)(\hat{x}), \hat{y} = f_r(\hat{x})}$$
$$\text{where} (\hat{x}, \hat{y}) \sim T_\theta(d, \hat{f}_b)$$

- **Code Accuracy.** To evaluate the utility of generated unit tests via code debugging, we follow prior work (Chen et al., 2023b; Chae et al., 2024) in reporting pass@1 code accuracy, i.e., the percentage of codes passing *all* unit tests after 3 rounds of debugging. Note that while we *debug* with model-generated UTs, we evaluate code accuracy using private *human-annotated* UTs.

## B    UTGEN Training

**Preprocessing Tulu Data.** We use the Tulu-3 SFT mixture dataset released by Lambert et al. (2024a) which contains a total of 939.3K instances.[6] However, it contains a mixture of prompts for instruction-following, math reasoning, and coding. Therefore, we filter for prompts involving Python coding by regex search for keywords "python" and "def " which suit our task setup described in Sec. 3.1. Furthermore, we filter out instances with more than 2K tokens in the prompt and ensure the prompt is a valid unicode string. This results in a total of 48.3K instances for which we use the "prompt" as the problem description and extract code from the response of the last turn when multi-turn interactions are present provided that the extracted code executes without any errors or issues. Finally, we prompt the LLM to be trained with UTGEN to generate 2 corrupted versions of this code to serve as the target code, and for each target code, we make 5 attempts to generate failing unit test inputs and filter out instances that do not have any such UTs. This is followed by the

---

[6]https://huggingface.co/datasets/allenai/tulu-3-sft-mixture

---

**Algorithm 2** `UTDebug`: Debugging with generated UTs

---

**Input:** $d, \hat{f}_b$ // *problem description and buggy code*
**Params:** Number of UTs $n$, Number of SC samples $k$, Unit Test Generator $T_\theta$, Number of debugging rounds $m$
**Output:** Debugged code $\hat{f}_e$
$i \leftarrow m$ // *Initializing number of rounds left*
$\hat{f}_e \leftarrow \hat{f}_b$ // *Initializing edited code with code to debug*
acceptEdit $\leftarrow$ *True* // *Accept edits to start debugging*
**while** $i > 0$ **do**
   $i \leftarrow i - 1$ // *One round of debugging*
   **if** acceptEdit $=$ *True* **then**
      $\mathcal{U} \leftarrow$ `BuildUT`$(d, \hat{f}_e)$ // *Obtain generated UTs*
      $(x_d, y_d) \leftarrow \varnothing$ // *Initialize UT for debugging feedback*
      **for** $(x, y) \in \mathcal{U}$ **do**
         **if** $\hat{f}_e(x) \neq y$ **then**
            $(x_d, y_d) \leftarrow (x, y)$ // *Failing UT to debug*
         prePass $\leftarrow$ `EvalCode`$(\hat{f}_e, \mathcal{U})$ // *Get pass rate*
         **if** $x_d = \varnothing$ **then** // *No need to debug*
            **return** $\hat{f}_e$
         **else** // *Prompt LLM to debug code*
            $f' \sim$ `LLM`$(\hat{f}_e | $`Debug`$(x_d, y_d, \hat{f}_e))$ // *Prompt LLM to debug code with UT feedback*
            postPass $\leftarrow$ `EvalCode`$(f', \mathcal{U})$
            **if** postPass $>$ prePass **then**
               $\hat{f}_e \leftarrow f'$ // *Based on validation on the generated UTs, accept the edits, otherwise discard, i.e., backtrack*
**return** $\hat{f}_e$

---

| Model | UT Method | HE+Fix | | | MBPP+Fix | | |
|---|---|---|---|---|---|---|---|
| | | Attack Rate | Out Acc. | Acc. ∩ Attack | Attack Rate | Out Acc. | Acc. ∩ Attack |
| Llama3 8B | Random | 89.63 | **72.97** | **72.97** | 62.56 | 41.85 | 24.28 |
| | Prompted | 95.73 | 39.59 | 38.78 | 62.67 | 29.64 | 16.41 |
| | UTGEN | **96.34** | 53.27 | 52.04 | **67.18** | **42.67** | **26.87** |
| Llama3.1 8B | Random | 76.02 | **63.52** | **63.52** | 47.28 | 36.0 | 21.33 |
| | Prompted | 92.68 | 34.53 | 33.89 | 59.28 | 19.38 | 11.08 |
| | UTGEN | **96.54** | 56.38 | 55.76 | **62.77** | **43.59** | **25.54** |
| Qwen2.5 7B | Random | 90.85 | **87.36** | **86.47** | 55.28 | **52.31** | 30.38 |
| | Prompted | 93.29 | 54.55 | 53.91 | 62.97 | 35.29 | 22.60 |
| | UTGEN | **96.54** | 72.90 | 72.48 | **65.54** | 48.62 | **32.82** |
| Qwen2.5 32B | Random | 84.96 | **80.19** | **79.95** | 54.56 | 50.17 | 34.87 |
| | Prompted | 88.82 | 64.77 | 63.89 | 66.87 | 35.59 | 27.38 |
| | UTGEN | **95.93** | 70.64 | 69.79 | **70.36** | 50.87 | **40.62** |

Table 6: Evaluation on intrinsic metrics on HE+Fix and MBPP+Fix for different UT generation methods across multiple models.

rationalization step (with the same LLM to be trained) described in Sec. 3.2 and in Fig. 2, which results in roughly 30K instances for each 7-8B model trained with UTGEN and nearly 70K instances for Qwen2.5 32B.

**Training Hyperparameters.** We train each model for 10 epochs with a batch size of 16 and a learning rate of 5e-6 with a cosine learning rate scheduler. Moreover, we only compute negative log-likelihood loss over the completions. We use LoRA (Hu et al., 2021) with a rank of 16, $\alpha = 32$ with a dropout of 0.05. We perform checkpoint selection based on the intrinsic

Acc. ∩ Attack metric. All training and inference is conducted on Nvidia's A6000 GPUs taking ≈20 GPU hours for training and < 0.5 GPU hours for inference (generating one code solution and corresponding UTs with test-time scaling). We train the larger Qwen2.5 32B model on 8 A100 GPUs for roughly 40 GPU hours using QLoRa (Dettmers et al., 2023) with 4-bit quantization.

## C  Overall Pipeline for UTDEBUG

In Algorithm 1 we describe the process of generating UTs for a candidate buggy code $\widehat{f}_b$ using any UT generation method and perform test time scaling for UT output prediction. This is then used within UTDEBUG as shown in Algorithm 2, which identifies failing UTs, debugs the target code $\widehat{f}_b$ based on feedback from failing UTs over multiple rounds, and returns the debugged (edited code). As illustrated in Fig. 3 (b), edits are accepted only if the newly generated code achieves a higher pass rate than the code before debugging, otherwise the edits are discarded.

**Inference Hyperparameters.**  When sampling multiple UTs and for generating LLMs response to UT feedback we use temperature sampling with temp = 0.7 and top_p = 0.9.

## D  Additional Intrinsic Evaluation

Similar to Table 1, we report intrinsic evaluation metrics (cf. Sec. 4) for HE+Fix and MBPP+Fix datasets in Table 6. Consistent with the findings in Sec. 5.1, without training, we observe a trade-off between attack rate and output accuracy, with randomly sampled UTs showing higher output accuracy whereas the prompted baseline exhibits higher attack rates and vice-versa. Once again UTGEN training best balances this trade-off yielding both high output accuracy and attack rate. However, due to the relative ease of attacking faulty codes in HE+Fix (initial pass rate of 0%) almost any generated UT fails and thus can be used for debugging. This, coupled with the higher output accuracy, results in the random baseline having the highest score on Acc. ∩ Attack. Note that we mitigate this impact on downstream performance by using test-time scaling for output prediction in UTDEBUG, which especially boosts the accuracies of targeted UTs generated based on the buggy code. On MBPP+Fix, UTGEN consistently yields the highest Acc. ∩ Attack score, followed by the random baseline.

To further validate our intrinsic evaluation metrics, specifically, that *both* high input attack rate and high output accuracy are key to downstream utility of model-generated unit tests – we measure the correlation between each property of the UT and whether the UT yielded successful debugging. To this end, we collect oracle metadata on all unit tests generated for MBPP+Fix with Qwen2.5 7B, i.e., we record if the unit test caused the target code to fail, whether its output was correct, and compute its Somers' D correlation with whether the target code was debugged successfully for $n = 1$ UTs and 1 round of debugging in Table 7. While individually, failing UTs and output correctness show mild correlation with downstream debugging, Table 7 shows that the presence of both attributes yields a strong (and the highest) correlation with debugging success and reinforces the motivation for UTGEN and improving unit-test generation ability of LLMs at large.

| Intrinsic Metric | Somers' Delta |
|---|---|
| Attack Rate | 0.43 |
| Output Acc. | 0.34 |
| Acc. ∩ Attack (Both) | 0.63 |

Table 7:  Somers' D correlation between intrinsic UT metrics and whether it resulted in successful debugging. Note, all reported results are statistically significant ($p < 0.05$).

## E  Additional Analysis and Results with UTDEBUG

**On Test-time Scaling and Backtracking.**  In Sec. 3.3, we highlighted the challenges of debugging with imperfect UTs and suggested remedies to make our debugging pipeline, UTDEBUG, robust to noisy feedback from such UTs. We study the effectiveness of these

measures, i.e., test-time scaling of UT outputs, and backtracking, on debugging with $n =$ 3 generated UTs for 3 rounds using Qwen2.5 7B on the MBPP+Fix dataset in Table 8.

We show that test-time scaling and back-tracking are *crucial* for LLM debugging with *generated* UTs. From Table 8, we observe that irrespective of the underlying method for UT generation (either randomly sampled or from UTGEN), removing either backtracking or test-time scaling hurts downstream performance. First, we find that removing test-time scaling for predicting the output of UTs decreases the performance of randomly-sampled UTs by 4% and that of UTGEN by 11.4%. Note that a larger drop in UTGEN's performance when removing test-time scaling of UT output

| UT Method | Acc. | Δ |
|---|---|---|
| Randomly-sampled (Qwen2.5 7B) | 34.77 | |
|    - Output Test-time Scaling | 30.77 | - 4.0 |
|    - Backtracking | 32.61 | - 2.2 |
| UTGEN (Qwen2.5 7B) | 37.54 | |
|    - Output Test-time Scaling | 26.15 | - 11.4 |
|    - Backtracking | 34.38 | - 3.2 |

Table 8: Ablating components of UTDEBUG's pipeline (cf. Sec. 3.3) for two different unit test generation methods (including UTGEN) on MBPP+Fix using Qwen2.5.

prediction is consistent with the findings in Sec. 5.1 that models struggle with reasoning over correct outputs for *failing UT inputs* (more often generated by UTGEN). Therefore, test-time scaling in output prediction provides a greater boost for reasoning over challenging inputs (Wang et al., 2022), and consequently, removing it yields larger drops for UTGEN. Moreover, Table 8 demonstrates that, without validation on other generated unit tests, LLMs tend to *overfit* on the unit test contained in the feedback, resulting in up to 3.2% drop in performance of UTGEN without backtracking.

**Comparison with CodeT (Chen et al., 2023a) for Debugging.** As mentioned in Sec. 2, UTGEN differs from Chen et al. (2023a) in that CodeT is designed for code generation by selecting a code from the largest consensus set of codes and independently generated UT (without conditioning on code). Alternatively, this UT sampling procedure is akin to the randomly-sampled baseline (described in Sec. 4) but the latter uses self-consistency for predicting the UT output. In contrast, UTGEN is designed for UT generation focusing on the *intrinsic quality of UTs* as well as their *utility for downstream tasks* like code-debugging and best-of-N ranking by conditioning

| Model | UT Method | HE+Fix | MBPP+Fix | (Hard) |
|---|---|---|---|---|
| | No UT | 52.53 | 23.08 | 16.47 |
| | Random | 79.75 | 34.77 | 17.06 |
| Qwen 2.5 7B | Prompted | 75.32 | 32.92 | 24.12 |
| | CodeT | 81.65 | 34.30 | 21.18 |
| | UTGEN | **82.91** | **37.54** | **29.41** |
| | No UT | 79.11 | 39.08 | 32.94 |
| | Random | 84.81 | 49.54 | 22.94 |
| Qwen 2.5 32B | Prompted | 85.44 | 50.77 | 40.59 |
| | CodeT | 86.71 | 46.61 | 29.41 |
| | UTGEN | **88.61** | **54.15** | **45.29** |

Table 9: Comparison of pass@1 accuracies after $n = 3$ rounds of debugging with UTDEBUG of different UT-generation methods including CodeT (Chen et al., 2023a) – a variant of random UT generation baseline – as well as UTGEN.

UT generation on the edge cases of a target code. To directly compare against CodeT, instead of using the code generated from CodeT, we use its unit tests in downstream debugging. In Table 9, we scale up CodeT such that it uses a similar computational budget as UTGEN and sample $n = 3$ unit tests for debugging with Qwen 2.5 7B and 32B Code-Instruct models. Across both models and all three debugging datasets, UTGEN outperforms debugging with unit tests generated by CodeT by as much as 7.54% (absolute) on MBPP+Fix and 15.88% (absolute) on MBPP+Fix (Hard) with Qwen 2.5 32B model. Moreover, CodeT (which generates UTs independent of code being debugged) lags behind the prompted UT baseline in nearly half of the settings, showing that conditioning UT generation on the erroneous code better helps identify and localize bugs.

# F   Scaling with Number of Generated UTs

Thus far, we use $n = 3$ generated UTs across baselines and models. However, as we described in Sec. 3.3, having multiple UTs can be advantageous because: (i) there is a higher likelihood

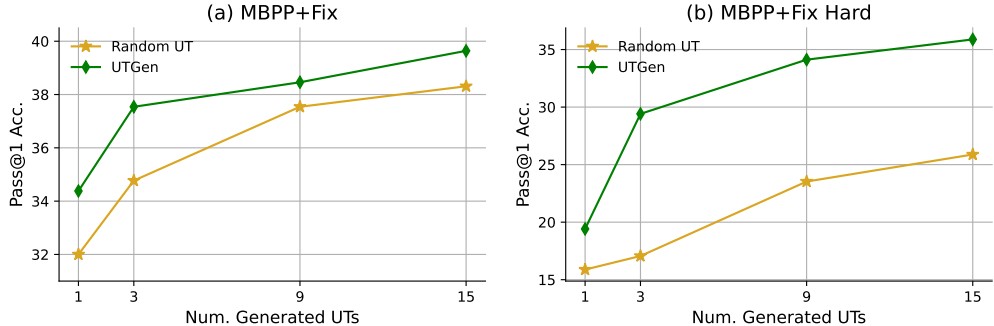

Figure 4: Increasing number of UTs across MBPP+Fix and MBPP+Fix (Hard) using UTs generated by UTGEN and randomly-sampling with Qwen2.5 7B.

of generating a failing UT and getting a reliable signal for when the code is correct, (ii) more robust signal for when to backtrack using validation on the entire generate test suite. We analyze the impact of increasing the number of generated UTs $n$ on downstream accuracy of Qwen2.5 7B after 3 rounds of debugging in Fig. 4. Our findings are as follows:

- First, Fig. 4 shows that despite increasing the number of generated unit tests, UTGEN consistently outperforms randomly sampled UTs that may not be failing. This highlights the benefits of generating *targeted unit tests conditioned on buggy code* in order to trigger errors and generate appropriate feedback for debugging.
- In settings with constrained resources, i.e., sampling $n \leq 3$ UTs, UTGEN is more effective at identifying errors by up to 3% on MBPP+Fix and 12% on MBPP+Fix (Hard).
- On MBPP+Fix (Hard), which contains less obvious errors and is harder to debug, we find that despite scaling to up to 15 generated UTs, the performance gap between UTGEN and randomly-sampled UTs remains at 10% (absolute).

## G Discussion

Our work on UTGEN contributes to the broader landscape of verification and feedback generation for LLM-generated code. While recent work has focused on training verifiers to provide feedback (Mahan et al., 2024; Zhang et al., 2024), a key challenge remains in obtaining high-quality feedback signals for debugging. UTGEN addresses this by directly generating unit tests that can identify problems in code, complementing existing work on how to effectively incorporate and present feedback for debugging (Chen et al., 2023b; Zhong et al., 2024) along with test-time scaling and backtracking incorporated in UTDEBUG. Our results demonstrate that without a quality signal to determine code correctness and/or how a faulty code is failing (in the form of unit tests), using LLMs to generate feedback and debug still proves to be challenging. This is one of the first efforts in this direction, and we hope to spark more interest in future work toward LLM-generated unit tests (both input and outputs) that reveal the model's coding errors.

Our approach connects to and complements recent work on handling real-world programming issues. While approaches designed for SWEBench (Jimenez et al., 2024) focus on fixing known issues from GitHub by understanding and implementing fixes for bug reports, UTGEN addresses the upstream challenge of automatically discovering potential issues in new code through test generation. Both tasks share a core challenge: determining the expected behavior of code without access to correct implementations. This connects to the fundamental concept of simulatability from computability theory (Rice, 1953), where we ask whether a program can predict the behavior of another program. Recent work such as Jain et al. (2024) shows that while LLMs can often simulate existing code by tracing execution steps, they struggle more with predicting correct outputs from specifications alone. Our results align with these findings – while UTGEN can generate test inputs that trigger errors (high attack rate), predicting correct expected outputs remains challenging (lower output

accuracy). This suggests that improving LLMs' ability to reason about intended program behavior from specifications remains a crucial direction for future work. Nevertheless, we find that the modifications made to debugging in UTDEBUG help boost UTGEN's accuracy and account for noise, leading to downstream gains.

## H  Prompts

In the following pages, we list all the prompts we use in this work.

---

**Prompted and UTGen Prompt for UT generation**

You are given a Python function {signature} to solve the following task:

## Task:

{description}

## Code Solution:

{code}

The code solution I have provided to you is **incorrect**. Your job is to give feedback by generating a unit test that

1. Is **valid** input based on the task description, i.e., an acceptable input consistent with task description that a correct program should be able to execute.

2. The output enclosed in . and is **faithful** to the task description, i.e., the output of the unit test is consistent with what a correct program would return.

3. **Breaks** the given code, i.e., does **not** execute to the **correct** output and brings out its mistakes and vulnerabilities.

Provide a reasoning for your answer and identify a general hypothesis or rationale identifying the cause of error. Then provide input and output of the unit test consistent with the pattern (hypotheis) you have identified. Note: - that you MUST directly write ALL input arguments of the function in the correct order. Skip writing any names of arguments. - you MUST enclose the unit test inputs and outputs in.

Respond in the format below:

## Hypothesis

<step-by-step reasoning >

Error Pattern: <an identified pattern of inputs that yields erroneous or incorrect outputs >

## Unit Test

### Input Arguments

<step-by-step reasoning for constructing a unit test that fits the error pattern identified above and is valid as per the task description >Arguments: {entry_point}(<all arguments >)

### Output

<step-by-step reasoning for what a **correct** {entry_point} would execute to based on the task description and your input above. Make sure your data type of the final answer matches the expected output type of the function. >

Output: <your final answer >

---

**No UT Feedback Prompt**

# Your Task

## Task {prompt}

## Code:

{code}

Based on given task and code, generate feedback that decides whether the code is correct or wrong in the format Feedback: <your feedback >. Always end your feedback with the line "The above code is correct." if it is correct, otherwise the feedback should end with "The above code is wrong, please fix it."

**UT Generation Prompt for Randomly-sampled UTs**

Given a Python function {signature} to solve the following task: {description}

Code Solution:

{code}

The code solution I have provided to you is incorrect. Your job is to give feedback by generating a unit test input that 1. Valid input based on the task description, i.e., an acceptable input consistent with task description that a correct program should be able to execute. 2. Given code will NOT be able to solve and brings out its mistakes and vulnerabilities.

Provide a reasoning for your answer and present your response in the format below:

<reasoning >

Arguments: {entry_point}(<all arguments >)

Note that you MUST directly write ALL input arguments of the function in the correct order. Skip writing any names of arguments.

**Training UT Gen – Code Corruption Prompt**

You are given a Python function {signature} to solve the following task:

## Task

{description}

## Correct Code Solution:

{code}

Assume you are a TA for a programming course. Your task is to corrupt the correct code or implementation of function {entry_point} to introduce realistic errors that can be made by your programming students. Note that you should write a code that fails one or more unit tests that this correct would succeed in. Also, give all your reasoning for why your generated code is incorrect outside the code block, i.e., **do not** leave comments in the code block that reveals the code is incorrect.

Give your output in the format:

<reasoning of error introduced >

## Incorrect Code Solution

<your generated incorrect code for {entry_point} >

**UT Feedback Prompt**

The above code is incorrect and does not pass the testcase.

Input: {wrong_testcase_input}

Output: {wrong_testcase_output}

Expected: {wrong_testcase_expected}

**Output Rationalization Prompt**

## Example

Given a Python function check(string) to solve the following task:

Write a python function to accept the strings which contains all vowels.

A user provides an the following input to function. The teacher lets you know that correct function generates the following output.

Input: "BCDEFG"

Output: False

Now **without** coding up the the function check(string), provide step-by-step reasoning for why function check when given input of "BCDFG" generates or returns False.

### Reasoning

Let's think step by step.

- According to the problem description, given a string the check should return accepted if it contains all the vowels and not accepted otherwise.

- The vowels (in lower case) that should be present in the string are: "a", "e", "i", "o" and "u".

- The given input is "BCDEFG" which contains characters: "b", "c", "d", "e", "f", "g".

- While the input string "BCDEFG" contains only vowel "e" and is missing vowels: "a", "i", "o", "u".

- Therefore, the output of the function is not accepted.

## Test Problem

Given a Python function {signature} to solve the following task:

{description}

A user provides an the following input to function {signature}. The teacher lets you know that correct function generates the following output.

Input: {unit_input}

Output: {unit_output}

Now **without** coding up the the function {signature}, provide step-by-step reasoning for why function {entry_point} when given input of {unit_input} generates or returns {unit_output}.

Write your output under header ### Reasoning.

