# OpenReview forum: "Learning to Generate Unit Tests for Automated Debugging"
_colmweb.org/COLM/2025/Conference — COLM 2025_

### Official Review · Reviewer_2H95 · 2025-04-28

**Rating:** 6
**Confidence:** 3
**Ethics Flag:** 1

**Summary:**

This paper attempts to enhance LLM-based program debugging by improving unit test generation. First, the authors proposed UTGen, a method for creating training data to finetune an LLM-based unit test (UT) generator. Three metrics were used to quantify the effectiveness of a unit test, including (1) the attack rate (i.e., whether the test case reveals the buggy behavior of the program) (2) the output accuracy (i.e., whether the generated output of the unit test is correct) and (3) whether the test case both has a correct output and can attack the program. Additionally, the authors proposed UTDebug, a debugging framework that is robust against noisy test cases. This includes using test-time scaling to improve output accuracy of the generated test cases and discarding edits if they do not improve the pass rate on the entire set of generated test cases. The experimental results show that test cases from UTGen had higher metric scores than baselines. They also worked well with UTDebug to fix buggy programs. Furthermore, there are experiments comparing UTGen with frontier LLMs and using unit tests for reranking generated coding solutions.

**Questions To Authors:**

**Questions**
1. In step 2(b) of Figure 2, can we use more than one test case when prompting the LLM to generate an edit? Why did you prefer using only one test case?
2. As the debugging is done iteratively, the order of the test cases used may impact the quality of the output program. How did you order the test cases in the experiments?
3. In Table 2, debugging with randomly-sampled UTs sometimes outperformed the prompted failing UTs. Why was this the case?
4. If we use a fine-tuned LLM or a frontier LLM to generate edits, will the UTs be less important?
5. It would be interesting to see the relationship between the intrinsic quality of the test cases and the effectiveness in the downstream debugging task. For instance, what is the difference between the pass@1 scores when the test case has a correct output versus when it has an incorrect output? Or when it can versus cannot attack the program? This will tell us how much the intrinsic evaluation metrics reflect the downstream debugging performance. Do you have any results along these lines?

**Suggestion**
- The last six lines on page 4 that discusses CoT reasoning seem related more to the training data generation step. So, it may be better to place them before the supervised finetuning paragraph.

**Reasons To Accept:**

- The finding about the tradeoff between attack rate and output accuracy is interesting, despite not being so surprising.
- Both UTGen and UTDebug performed well in the experiments.
- Using unit tests for reranking generated codes is a good application of UTGen.

**Reasons To Reject:**

- There is no experiment evaluating the proposed UTDebug framework specifically. We do not know how effective are the two strategies (i.e., test-time scaling and back-tracking), which are pretty computational intensive.
- The set of generated test cases is used to decide whether to apply an edit. However, this set is unlikely to cover test cases that the original program passes since the fine-tuned UT generator focuses on test cases with high attack rate. Therefore, an approved edit might make the program lose its original correct behavior on ungenerated test cases.
- The writing and presentation could be improved.
   - The paper is not self-contained as several important details are in the appendix. For example, although Figure 3 is in the Appendix, it is referred to many times in Section 3.2. In my opinion, it would be better to place Figure 3 in the main content as it helps readers understand the UTGen pipeline.
   - In contrast, the introduction could be more concise as the current form may overload the readers with details that are difficult to understand until they read the following sections.
   - The paper mentions “LLMs” several times, but it is unclear whether each mention refers to the LLM used for generating test cases or the LLM used for generating edits (or they are the same model).

---

> ### Author Response · Authors · 2025-05-31
> **Response to Reviewer 2H95 (Part 1/2)**
>
> We thank you for your detailed comments and questions. We attempt to address each of them in detail below:
>
> > **\[R1\] There is no experiment evaluating the proposed UTDebug framework specifically**
>
> Due to the space constraints, we moved this analysis to Appendix E and Table 6 where we ablate both strategies test-time scaling and backtracking one at a time on Qwen 2.5 7B using both randomly-generated UTs and those by UTGen. On average we find that without test-time scaling, debugging performance on MBPP+Fix decreases by 7.7% (absolute). Similarly, removing backtracking in UTDebug yields an average drop of 2.7% in debugging performance. While we point to this in footnote 4 of the main paper, we will use the extra page in the final submission to expand on this.
>
> ---
>
> > **\[R2\] … an approved edit might make the program lose its original correct behavior on ungenerated test cases.**
>
> Your point is well taken, however, based on the empirical results we make the following arguments:
>
> - When conditioning the UT-generator on the target code (as in UTGen and prompted baseline), the model is tasked with figuring out challenging edge cases to test the code’s correctness. The UT output is consistent with the task description, meaning that *once generated, a UT can be used for validating the quality of edits and backtracking (L217 \- 227\) even in subsequent rounds, even if the target code has now been edited*. This ensures the edits are more robust in UTDebug as the number of rounds increases.
> - Second, if conditioning UTs on a target code was suboptimal for debugging, the random baseline (without any such conditioning) would outperform UTGen. However, *we find that UTGen consistently outperforms the random baseline for multiple model families and scales*. Additionally, the intrinsic evaluation shows that UTs generated by *UTGen are better at predicting if a code is incorrect (by attack rate)* and we reaffirm this finding in response to Q5 below.
> - Lastly, we agree that increasing the number of UTs would improve the overall debugging process as it provides a higher chance for any errors to be revealed. We test this in Appendix F, where we scale the number of UTs ($n$) generated in each round from 1 to 15 in Fig 4, finding that debugging performance generally scales with increasing $n$. However, we note that the ***improvements from UTGen over baselines persist even as we increase n=15 unit tests in each round, with an improvement of \+10% (absolute) over randomly-sample UTs on MBPP+Fix (Hard)***.
>
>
>
> ---
>
> > **\[Q1\] Can we use more than one test case when prompting the LLM to generate an edit?**
>
> In our preliminary experiments, we found focussing on *one failing UT at a time to be just as good if not slightly better than using feedback from all failing unit tests* while requiring fewer input tokens. One possible explanation is that due to the **multi-round debugging framework** we use. Once the code is edited to resolve one failing UTs, in the next round the model generates more challenging UTs conditioned on the edited code, and can **continue to debug** if there are any errors or failures. Note that using feedback from one failing UT is consistent with prior debugging works such as Chen et al., (2023b) and Zhong et al., (2024).
>
> ---
>
> > **\[Q2\] How did you order the test cases in the experiments?**
>
> In cases where the model generates multiple failing UT, we select the unit test mentioned in the feedback prompt *randomly*. However, as we explained above, feedback from other failing UTs is ***indirectly** utilized in the form of pass rate to determine if an edit should be accepted or rolled back*.
>
> ---
>
> > **\[Q3\] In Table 2, debugging with randomly-sampled UTs sometimes outperformed the prompted failing UTs. Why was this the case?**
>
> As mentioned in Sec 3.3, debugging with model-generated UTs can result in noisy feedback, for e.g., if the output of UT is predicted incorrectly. This, coupled with the findings in Sec 5.1 (L308-314) that models struggle to correctly reason over the output of more challenging UT inputs, could explain your observation. Specifically, in the random baseline, the UTs generated are not conditioned on the target code and therefore, may correspond to *relatively simpler inputs whose outputs are easier to predict*. On the other hand, in the prompted baseline, the model might generate more challenging UT inputs conditioned on the edge-cases of the target code; if *it fails to generate the correct output, the resulting debugging feedback would be less useful*. This motivates the need for training LLMs (as we do in UTGen) to generate ***both failing inputs and the correct UT outputs.*** These findings are also corroborated by the intrinsic metrics on HE+Fix and MBPP+Fix datasets in Appendix D and Table 5, where the random baseline scores are higher than the prompted baseline.

---

> > ### Author Response · Authors · 2025-05-31
> > **Response to Reviewer 2H95 (Part 2/2)**
> >
> > > **\[Q4\] If we use a fine-tuned LLM or a frontier LLM to generate edits, will the UTs be less important?**
> >
> > We thank you for this suggestion and conduct a new experiment in which we use GPT-4o (a frontier model) as the debugger on MBPP+Fix (Hard). We compare “no UT” feedback generated by GPT-4o as well as the smaller Qwen 32B model with UT-feedback-based debugging with the prompted baseline and UTGen. The results are as follows:
> >
> > | Debugger Model | UT Method (Model) | MBPP+Fix (Hard) |
> > | :---- | :---- | :---- |
> > | GPT4o | No UT (self) | 34.71 |
> > | GPT4o | No UT (Qwen 32B) | 32.94 |
> > | GPT4o | Prompted (Qwen 32B) | 45.88 |
> > | GPT4o | UTGen (Qwen 32B) | **59.69** |
> >
> > - Even with a stronger debugging model (GPT-4o), debugging based on feedback from ***UTGen is the most effective even when the UTs are generated by a smaller Qwen 32B model***.
> > - Both the prompted baseline and UTGen outperform the No UT baselines by 11.17% and \~25% (absolute) respectively. This reiterates that even with a stronger debugger, *UT feedback is important*.
> > - Lastly, we can conclude that the *quality of UTs remains to be important*, as debugging with UTGen outperforms the prompted baseline with lower intrinsic scores by 13.8% (absolute).
> >
> > ---
> >
> > > **\[Q5\] It would be interesting to see the relationship between the intrinsic quality of the test cases and the effectiveness in the downstream debugging task. … Do you have any results along these lines?**
> >
> > To clarify, Sec 5.1, 5.2 and 5.3 together show that unit tests generated by UTGen have the highest intrinsic scores as well as yield the best downstream performance (measured by code-debugging and best-of-N sampling). Based on your question, we collect oracle metadata on all unit tests generated for MBPP+Fix with Qwen2.5 7B. Specifically, we check if the unit test caused the target code to fail, whether its output was correct, and compute its Somer’s D correlation with whether the target code was debugged successfully for n=1 UTs and 1 round of debugging. *While individually, failing UTs and output correctness show mild correlation with debuggability, we find that the **presence of both attributes yields a strong (and the highest) correlation with debugging success**.* Note that all correlations yielded p \< 0.05 and are statistically significant. We will incorporate these results in the final version.
> >
> > | Intrinsic Metric | Somer’s D correlation with successful debugging |
> > | :---- | :---- |
> > | Failing UTs (Attack Rate) | 0.43 |
> > | Correct UT output (Output Acc.) | 0.34 |
> > | Both (Acc. ∩ Attack) | **0.63** |
> >
> > ---
> > ---
> >
> > Finally, we thank you for your suggestions on improving the organization and presentation of the paper and we will incorporate these in the final version of our paper. We hope our response has addressed all of your questions and will allow you to revisit your score. We are happy to answer any remaining or follow up questions/requests you may have.

---

> > > ### Author Response · Authors · 2025-06-05
> > > **Gentle Reminder**
> > >
> > > Since the response period is underway, we wanted to check in and see if our response has addressed your comments and will allow you to revisit your score — otherwise we’re happy to continue discussing in the remaining days of the rebuttal period.

---

> > ### Comment · Reviewer_2H95 · 2025-06-09
> >
> > Thank you for responding to my comments and for providing the additional results. I found several points interesting, particularly your answers to [Q4] and [Q5]. Overall, I'm going to increase my score from 5 to 6. I hope you incorporate the author response into the next version of your paper.
> >
> > Regarding [R2], I'd like to clarify my point a bit. I acknowledge the effectiveness of UTGen, and I do not expect the random baseline to outperform UTGen in the experiment. However, I still see the benefit of test cases that the original program passes. Although such test cases may not be very helpful when generating edits, they are likely helpful when calculating the pass rate to determine if an edit should be accepted or rolled back.

---

> > ### Author Response · Authors · 2025-06-09
> >
> > Thank you for appreciating the new results and clarifying your point. Additionally, in response to R2, in the first point, we clarify that UTDebug already incorporates this in the debugging design. At iteration 1, if the model generates 3 UTs, out of which 2 are failing and 1 passes, we retain ***all 3 UTs***. At the end of round 1, edits are incorporated if one of the failing UTs is fixed; in the next round, 3 fresh UTs are generated, but the backtracking and validation is done based on ***all 6 UTs generated so far***. Furthermore, your point is corroborated by the code-generation results, which show ranking multiple generations based on pass rate from a batch of UTs generated from UTGen is the *"best judge"* of code correctness.
> >
> > Please let us know if you have any remaining comments or questions that would strengthen the submission in your view and help you revisit your score/*increase it further*. We will certainly incorporate these additional points in the final version of the paper.

---

### Official Review · Reviewer_h2jQ · 2025-05-12

**Rating:** 6
**Confidence:** 4
**Ethics Flag:** 1

**Summary:**

**Motivation:**

The paper addresses the challenge of generating automated unit tests for debugging faulty code generated by large language models (LLMs). Unit tests are crucial for verifying code correctness and guiding debugging, but existing methods often struggle to generate both error-revealing inputs and accurate outputs without access to the correct solution. The authors identify a key trade-off between generating unit test inputs that detect errors and correctly predicting the unit test output.

**Methods:**

To overcome this trade-off, the authors propose **UTGEN**, a framework that teaches LLMs to generate unit test inputs that reveal errors along with their correct expected outputs. Since LLM-generated tests can be noisy (e.g., incorrect outputs), they also introduce **UTDEBUG**, which scales UTGEN through test-time compute for better output prediction and applies a validation and backtracking mechanism to improve debugging accuracy. UTGEN is trained on code generation datasets by introducing controlled errors, and UTDEBUG improves test accuracy by generating multiple tests and retaining edits that consistently improve pass rates.

**Datasets/Benchmarks:**

The authors introduced UTGEN, a dataset created by perturbing reference code to simulate faulty scenarios, thereby generating input-output pairs for training and evaluating unit test generation. Additionally, they developed a debugging split of MBPP+  called **MBPP+Fix (Hard)**, which contains more challenging errors by filtering faulty codes that pass between 50% and 95% of unit tests. This dataset emphasizes subtle and logically complex errors, making it more difficult for models to debug compared to existing datasets.

**Reasons To Accept:**

**Quality and Clarity:**

* The paper is well-written, clearly structured, and provides comprehensive details regarding methodology design, dataset creation, and experimental procedures.

**Novelty and Soundness:**

* The paper addresses a novel research question: the inherent trade-off between the attack rate of unit tests and their correctness.
* The development of new training datasets for unit test generation addresses a pressing gap in the current research landscape, providing resources that can facilitate further research in this domain.
* The introduction of the **MBPP-Fix (Hard)** benchmark is helpful, considering the nearing saturation of model performance on existing benchmarks like HumanEvalFix.
* The experimental evaluation is thorough, exploring multiple downstream applications of generated unit tests. The authors provide a comprehensive analysis and discussion of results.

**Reasons To Reject:**

**Methodology Design:**

1. **Insufficient Justification of CoT and Reject Sampling:**

   * The paper leverages Chain-of-Thought (CoT) prompting and reject sampling for dataset generation. However, these techniques are not novel in the context of dataset generation. The paper lacks a deeper analysis or discussion on how these methods fundamentally address the identified trade-off between attack rate and output correctness in unit test generation. Providing concrete evidence or theoretical analysis to substantiate the choice of these techniques would strengthen the methodological soundness.

2. **Evaluation Scope and Baselines:**

   * The current evaluation framework focuses primarily on baseline comparisons involving basic code generation/debugging models. However, several existing works (e.g., CodeT and others) also employ self-consistency or similar techniques for generating unit tests in HumanEval/MBPP-like coding problems. To convincingly demonstrate the effectiveness of UTGEN, the evaluation should be expanded to include more advanced baselines that also utilize generated unit tests for code generation or debugging. This would provide a more comprehensive and rigorous assessment of UTGEN’s relative strengths and limitations.

---

> ### Author Response · Authors · 2025-05-31
> **Response to Reviewer h2jQ**
>
> We thank the reviewer for their helpful comments and feedback, as well as for acknowledging the paper to be “clearly structured”, “addressing a novel research question”, introducing a “helpful benchmark”, and the  experimental evaluation to be “thorough” and “comprehensive”.
>
> One of our core contributions lies in characterizing the generation of failing or challenging unit test inputs and predicting correct outputs for them as **reasoning tasks for coding LLMs**, which motivates our use of chain-of-thought strategies (Wei et al 2022). Please see our detailed responses below:
>
> > **\[W1\] Insufficient Justification of CoT and Reject Sampling**
>
> We would like to clarify that our research contributions include *identifying the tradeoff in unit test generation via intrinsic properties* and *designing a training pipeline that bootstraps* necessary data for training LLMs to be better unit test generators *from code-generation datasets*. We acknowledge that the broader use  of chain of thought or rejection sampling are not novel and have been used in prior work (including data generation) – therefore, we ***do not claim these as contributions of our work***. Additionally, we use *rejection sampling for curating the training data* because prior to training, models do not always generate failing unit test inputs, thus, requiring additional filtering. We will expand on this discussion in the final version of our paper.
>
> ---
>
> > **\[W2\] Evaluation scope and baselines: Comparison with CodeT baseline**
>
> UTGen differs from CodeT in that CodeT is designed for code generation by selecting a code from the largest consensus set of codes and independently generated UT (*without conditioning on code*). In contrast, UTGen is designed for UT generation focussing on the ***intrinsic quality of UTs*** as well as their ***utility for downstream tasks*** like code-debugging and best-of-n ranking by *conditioning UT generation on the edge cases of a target code*. To directly compare against CodeT, instead of using the code generated from CodeT, we use its unit tests in downstream debugging (L114-117). We would also like to point out that the UT generation procedure in the randomly sampled baseline (L269-272) is consistent with the UT generation procedure in CodeT, but  uses self-consistency for predicting the output of the UT.
>
> We run this experiment by scaling up CodeT such that it uses a similar computational budget as UTGen and sample n=3 unit tests for debugging with Qwen 2.5 7B and 32B Code-Instruct models.
>
> | Model | UT Method | HE+Fix | MBPP+Fix | (Hard) |
> | :---- | :---- | :---- | :---- | :---- |
> | Qwen 2.5 7B | No UT | 52.53 | 23.08 | 16.47 |
> |  | Random  | 79.75 | 34.77 | 17.06 |
> |  | Prompted | 75.32 | 32.92 | 24.12 |
> |  | CodeT | 81.65 | 34.30 | 21.18 |
> |  | **UTGen** | **82.91** | **37.54** | **29.41** |
> | Qwen 2.5 32B | No UT | 79.11 | 39.08 | 32.94 |
> |  | Random  | 84.81 | 49.54 | 22.94 |
> |  | Prompted | 85.44 | 50.77 | 40.59 |
> |  | CodeT | 86.71 | 46.61 | 29.41 |
> |  | **UTGen** | **88.61** | **54.15** | **45.29** |
>
> **Across both models and all three debugging datasets, UTGen outperforms debugging with unit tests generated by CodeT by as much as 7.54% (absolute) on MBPP+Fix and 15.88% (absolute) on MBPP+Fix (Hard) with Qwen 2.5 32B model.** Moreover, CodeT (which generates UTs independent of code being debugged) lags behind the prompted UT baseline in 3/6 settings, showing that conditioning UT generation on the erroneous code better helps identify and localize bugs. We find CodeT performs slightly better than the randomly-sampled UT baseline (especially for MBPP+Fix Hard split). We will include these results in the final version of the paper.
>
> ---
>
> We hope to have addressed your questions satisfactorily, and will be happy to engage in further discussions based on any follow up questions you may have.

---

> > ### Comment · Reviewer_h2jQ · 2025-06-07
> >
> > Thank you for the clarification and the additional results. I’ll stick with my current score for now.

---

> > > ### Author Response · Authors · 2025-06-07
> > > **Follow-up to response by h2jQ with additional results**
> > >
> > > Thank you for your positive review and feedback. Our rebuttal has shown that UTGen outperforms debugging with unit tests generated by CodeT by as much as 7.54% (absolute) on MBPP+Fix and 15.88% (absolute) on MBPP+Fix (Hard) with Qwen 2.5 32B model.
> > >
> > > Additionally, we would point out **two additional positive results** from experiments run in the rebuttal period showing that (i) UTGen helps improve downstream debugging even with a strong frontier model as a debugger; and (ii) both our intrinsic metrics jointly correlate best and strongly with the ability to debug.
> > >
> > > > **UTGen improves debugging with a frontier model as debugger**
> > >
> > > We conduct a new experiment in which we use GPT-4o (a frontier model) as the debugger on MBPP+Fix (Hard). We compare “no UT” feedback generated by GPT-4o as well as the smaller Qwen 32B model with UT-feedback-based debugging with the prompted baseline and UTGen. The results are as follows:
> > >
> > > | Debugger Model | UT Method (Model) | MBPP+Fix (Hard) |
> > > | :---- | :---- | :---- |
> > > | GPT4o | No UT (self) | 34.71 |
> > > | GPT4o | No UT (Qwen 32B) | 32.94 |
> > > | GPT4o | Prompted (Qwen 32B) | 45.88 |
> > > | GPT4o | UTGen (Qwen 32B) | **59.69** |
> > >
> > > - Even with a stronger debugging model (GPT-4o), debugging based on feedback from ***UTGen is the most effective even when the UTs are generated by a smaller Qwen 32B model***.
> > > - Both the prompted baseline and UTGen outperform the No UT baselines by 11.17% and \~25% (absolute) respectively. This reiterates that even with a stronger debugger, *UT feedback is important*.
> > > - Lastly, we can conclude that the *quality of UTs remains to be important*, as debugging with UTGen outperforms the prompted baseline with lower intrinsic scores by 13.8% (absolute).
> > >
> > > ---
> > >
> > > > **Intrinsic metrics of test case quality correlate with downstream debugging success**
> > >
> > > To test how well our intrinsic metrics correlate with downstream debugging success, we collect oracle metadata on all unit tests generated for MBPP+Fix with Qwen2.5 7B. Specifically, we check if the unit test caused the target code to fail, whether its output was correct, and compute its Somer’s D correlation with whether the target code was debugged successfully for n=1 UTs and 1 round of debugging. *While individually, failing UTs and output correctness show mild correlation with debuggability, we find that the **presence of both attributes yields a strong (and the highest) correlation with debugging success**.* Note that all correlations yielded p \< 0.05 and are statistically significant. We will incorporate these results in the final version.
> > >
> > > | Intrinsic Metric | Somer’s D correlation with successful debugging |
> > > | :---- | :---- |
> > > | Failing UTs (Attack Rate) | 0.43 |
> > > | Correct UT output (Output Acc.) | 0.34 |
> > > | Both (Acc. ∩ Attack) | **0.63** |
> > >
> > > ---
> > >
> > > In light of these positive results motivating the importance and effectiveness of UTGen, we would kindly request you to reconsider raising your overall review and score of our work.

---

> > > > ### Author Response · Authors · 2025-06-09
> > > > **Friendly follow-up reminder for Reviewer h2jQ**
> > > >
> > > > We wanted to politely check in to see if you had the chance to review the additional experiments (**on UTGen improving frontier model debuggers and correlation of intrinsic metrics and downstream performance**) we shared and which other reviewers found quite interesting. We are happy to discuss them in more detail in the remaining 1 day of the rebuttal period, if you have any comments or suggestions that would allow you to reconsider revisiting your ratings.

---

### Official Review · Reviewer_6enx · 2025-05-13

**Rating:** 5
**Confidence:** 4
**Ethics Flag:** 1

**Summary:**

The paper introduces UTGEN, a framework that trains LLMs generate high-quality unit tests aimed at identifying faults in buggy code without requiring gold-standard implementations. The authors demonstrate a key trade-off between generating test cases that reveal errors (high attack rate) and those that produce correct expected outputs (output accuracy). UTGEN addresses this trade-off by training LLMs with synthetic training data derived from perturbed code and reference outputs. Secondly, the authors introduce UTDEBUG, a debugging framework that leverages the generated UTs for iterative code correction. UTDEBUG incorporates inference-time self-consistency (test-time scaling) and backtracking mechanisms to mitigate the noise in UT feedback. The authors evaluate the system across several benchmarks (e.g., HumanEvalFix and MBPP+Fix), showing improved debugging performance.

**Questions To Authors:**

- What is the main differences between current approach (UTGEN, UTDEBUG) and CodeT? Can you compare their performance by using the same inference config?
- How good is the perturbed code created by LLM? Any human study to determine the quality of the dataset?
- While methods like CodeT can be expensive due to large-scaling test case generation, the computation cost might be cheaper than training a UT generator with UTGEN + test-time scaling with UTDEBUG. Do you have any analysis of the compute cost of the current approach against the baselines like CodeT?

**Reasons To Accept:**

- The paper highlights a trade-off in UT generation (attack rate vs. output accuracy) and proposes a data generation and training pipeline (UTGEN) to mitigate it. However, I would say a related work (CodeT) already indirectly addressed this trade-off by generating large-scale test cases and statistically determine better code candidates based on the clusters of candidates grouped by test outputs and the corresponding sizes of the clusters. How is the current approach better than CodeT?
- Comprehensive Evaluation with different metrics to evaluate the quality of UTs e.g. attack rate, output accuracy, passing rate of generated code
- The authors apply on different Open-source models (Llama3, Qwen) from 8B to 32B, offering a cost-effective alternative to commercial LLMs.

**Reasons To Reject:**

- In the current method, the reliance on artificially perturbed code may limit generalization to organically occurring bugs in real-world repositories. Furthermore, the perturbation is done via LLM and there is not much description of the perturbation method and how realistic the perturbed code is? The authors should at least perform a human study on a subset of data.
- While UTDEBUG provides a good strategy to handle noisy feedback via CoT+self-consistency and backtracking, the test-time self-consistency and multi-round debugging can be computationally expensive. Note that this computation cost is incurred on top of the cost to train UTGEN to generate test cases.
- Limited Comparisons with stronger UT Generator baselines: While LLM baselines are well-covered, comparison with stronger baselines such as CodeT is missing.

---

> ### Author Response · Authors · 2025-05-31
> **Response to Reviewer 6enx**
>
> We thank you for your insightful questions and feedback. We are glad that you appreciated the “tradeoff in UT generation” highlighted by our work, “comprehensive evaluation” and that UTGen offers a “cost-effective alternative to commercial LLMs”. We respond to your comments and queries below:
>
> ---
>
> > **\[R3, Q1\] How is the current approach better than CodeT?**
>
> UTGen differs from CodeT in that CodeT is designed for code generation by selecting a code from the largest consensus set of codes and independently generated UT (*without conditioning on code*). In contrast, UTGen is designed for UT generation focussing on the ***intrinsic quality of UTs*** as well as their ***utility for downstream tasks*** like code-debugging and best-of-n ranking by *conditioning UT generation on the edge cases of a target code*. To directly compare against CodeT, instead of using the code generated from CodeT, we use its unit tests in downstream debugging (L114-117). We would also like to point out that the UT generation procedure in the randomly sampled baseline (L269-272) is consistent with the UT generation procedure in CodeT, but  uses self-consistency for predicting the output of the UT.
>
> We run this experiment by scaling up CodeT such that it uses a similar computational budget as UTGen and sample n=3 unit tests for debugging with Qwen 2.5 7B and 32B Code-Instruct models.
>
> | Model | UT Method | HE+Fix | MBPP+Fix | (Hard) |
> | :---- | :---- | :---- | :---- | :---- |
> | Qwen 2.5 7B | No UT | 52.53 | 23.08 | 16.47 |
> |  | Random  | 79.75 | 34.77 | 17.06 |
> |  | Prompted | 75.32 | 32.92 | 24.12 |
> |  | CodeT | 81.65 | 34.30 | 21.18 |
> |  | **UTGen** | **82.91** | **37.54** | **29.41** |
> | Qwen 2.5 32B | No UT | 79.11 | 39.08 | 32.94 |
> |  | Random  | 84.81 | 49.54 | 22.94 |
> |  | Prompted | 85.44 | 50.77 | 40.59 |
> |  | CodeT | 86.71 | 46.61 | 29.41 |
> |  | **UTGen** | **88.61** | **54.15** | **45.29** |
>
> **Across both models and all three debugging datasets, UTGen outperforms debugging with unit tests generated by CodeT by as much as 7.54% (absolute) on MBPP+Fix and 15.88% (absolute) on MBPP+Fix (Hard) with Qwen 2.5 32B model.** Moreover, CodeT (which generates UTs independent of code being debugged) lags behind the prompted UT baseline in 3/6 settings, showing that conditioning UT generation on the erroneous code better helps identify and localize bugs. We find CodeT performs slightly better than the randomly-sampled UT baseline (especially for MBPP+Fix Hard split). We will include these results in the final version of the paper.
>
> ---
>
> > **\[R1, Q2\] How good is the perturbed code created by LLM? Any human study to determine the quality of the dataset?**
>
> We clarify that our evaluation datasets (HE+Fix, MBPP+Fix) f*eature human-written or realistic code-generation errors from smaller LMs, not artificially-perturbed code*  (L237-250, Appendix A.1)*.* LLM perturbation of gold codes was exclusively used for UTGen training due to limited dedicated datasets (L161-165) and we expand on the training setup in Appendix B (L693-707) as well as the perturbation on Page 21 (bottom). UTGen's demonstrated efficacy in code-debugging and Best-of-N ranking (Sec 5.2 and 5.3), even with initial filtering to control the quality of the perturbed code (L701-704), confirms the high quality of our training data.
>
> ---
>
> > **\[R2, Q3\] Do you have any analysis of the compute cost of the current approach against the baselines like CodeT?**
>
> As explained in L276-277, all unit test generation approaches use the test-time scaling and backtracking features of UTDebug, making them computationally comparable, i.e., similar number of LLM calls, token budgets and run-times. Based on our response to Q1/R3 above, we scale up CodeT’s unit test generation capabilities to also have a comparable computational budget as well as a similar number of UTs sampled.  Additionally, we point out that we train UTGen models with a standard academic budget (L708-716), *outperforming other trained counterparts like external state-of-the-art 8B RM (Liu et al 2024a) and yielding **similar if not better** UT-generation capabilities compared **to** **frontier LLMs that are monetarily more expensive**.* Lastly, we plan on **publicly releasing** our trained models for the research community to use directly.
>
> ---
>
> We hope our response has addressed all of your questions and will allow you to revisit your score. We are happy to answer any followup questions and requests you may have.

---

> > ### Author Response · Authors · 2025-06-05
> > **Gentle Reminder**
> >
> > Given that the response period is now underway, we wanted to check in and see if our response has addressed your comments and will allow you to revisit your score — otherwise we’re happy to continue discussing in the remaining days of the rebuttal period.

---

> > > ### Comment · Reviewer_6enx · 2025-06-07
> > >
> > > Thanks for your response. I decided to raise my rating.
> > >
> > > Regards,

---

> > > > ### Author Response · Authors · 2025-06-08
> > > >
> > > > Thank you for your response and for raising your score as a result of our additional results and explanations. By the way, we noticed that your rating is still "Marginally below acceptance threshold" -- so is there anything else we can answer/share in the remaining discussion period that might further increase your assessment? We are more than happy to engage with you further during the remainder of the discussion period to address any remaining comments/questions. Thanks again!

---

> > ### Author Response · Authors · 2025-06-10
> > **Final Reminder and Additional Results**
> >
> > Thank you for your positive review and feedback.  Our rebuttal has shown that UTGen outperforms debugging with unit tests generated by CodeT by as much as 7.54% (absolute) on MBPP+Fix and 15.88% (absolute) on MBPP+Fix (Hard) with Qwen 2.5 32B model. Given that today is the last day of the discussion period, we would like to draw your attention to **additional positive results** from experiments we ran in the rebuttal period showing that ***UTGen helps improve downstream debugging even with a strong frontier model as a debugger***.
> >
> > > **UTGen improves debugging with a frontier model as debugger**
> >
> > We conduct a new experiment in which we use GPT-4o (a frontier model) as the debugger on MBPP+Fix (Hard). We compare “no UT” feedback generated by GPT-4o as well as the smaller Qwen 32B model with UT-feedback-based debugging with the prompted baseline and UTGen. The results are as follows:
> >
> > | Debugger Model | UT Method (Model) | MBPP+Fix (Hard) |
> > | :---- | :---- | :---- |
> > | GPT4o | No UT (self) | 34.71 |
> > | GPT4o | No UT (Qwen 32B) | 32.94 |
> > | GPT4o | Prompted (Qwen 32B) | 45.88 |
> > | GPT4o | UTGen (Qwen 32B) | **59.69** |
> >
> > Even with a stronger debugging model (GPT-4o), debugging based on feedback from ***UTGen is the most effective even when the UTs are generated by a smaller Qwen 32B model***, outperforming the second-best, prompted baseline with lower intrinsic scores by 13.8% (absolute).
> >
> > ---
> >
> > In light of these positive results motivating the importance and effectiveness of UTGen, we would kindly request you to reconsider raising your overall review and score of our work.

---

### Decision · Program_Chairs · 2025-07-08

**Decision:**

Accept

**Comment:**

This paper presents an approach of unit test generation, and a corresponding pipeline UTDebug to utilize LLM-generated unit tests for debugging. The challenges discussed from this work include whether the generated unit tests can effectively reflect code errors, and whether the test output prediction is accurate. In the evaluation, the paper presents an analysis on the quality of generated unit tests with quantitative metrics to reflect the attack rate and output prediction accuracy. Meanwhile, UTDebug improves the performance and outperforms baselines such as CodeT.

In the rebuttal, the authors added important new experiments to show the effectiveness of their approach, including the comparison to CodeT, and results using the GPT-4o model for debugging. The new analysis and experiments addressed main concerns of the initial submission. The authors should incorporate these changes in the final version.